# A non-linear data driven approach to bias correction of XCO2 for OCO-2 NASA ACOS version 10

William Keely[1], Steffen Mauceri[2], Sean Crowell[3], Christopher O'Dell[4]

[1]Data Science and Analytics Institute, University of Oklahoma, Norman, OK, USA
[2]Jet Propulsion Laboratory, California Institute of Technology, Pasadena, CA, USA
[3]School of Meteorology, University of Oklahoma, Norman, OK, USA
[4]Cooperative Institute for Research in the Atmosphere, Colorado State University, Fort Collins, CO, USA

*Correspondence to*: William Keely (william.r.keely@ou.edu)

**Abstract.**

Measurements of column averaged dry air mole fraction of $CO_2$ (termed $XCO_2$) from the Orbiting Carbon Observatory-2 (OCO-2) contain systematic errors and regional scale biases; often induced by forward model error or nonlinearity in the retrieval. Operationally, these biases are corrected for by a multiple linear regression model fit to co-retrieved variables that are highly correlated with $XCO_2$ error. The operational bias correction is fit in tandem with a hand-tuned quality filter which limits error variance and reduces the regime of interaction between state variables and error to one that is largely linear. While the operational correction and filter are successful in reducing biases in retrievals, they do not allow for throughput or correction of data in which biases become nonlinear in predictors or features. In this paper, we demonstrate a clear improvement in the reduction of error variance over the operational correction by using a set of non-linear machine learning models, one for land and one for ocean soundings. We further illustrate how the operational quality filter can be relaxed when used in conjunction with a non-linear bias correction, which allows for an increase of sounding throughput by 14% while maintaining the residual error of the operational correction. The method can readily be applied to future ACOS algorithm updates, OCO-2's companion instrument OCO-3, and to other retrieved atmospheric state variables of interest.

## 1 Introduction

Carbon dioxide ($CO_2$) is a key contributor to radiative forcing and hence, rising levels in the atmosphere are of concern due to their influence on future climate. Following a long history of critical in situ measurements of $CO_2$ at key sites around the world that allowed us to better understand the carbon cycle on continental scales, the era of space-based remote sensing began with the Scanning Imaging Absorption spectrometer for Atmospheric Chartography (SCIAMACHY) in March 2002 (Bovensmann et al, 1999) and the Atmospheric Infrared Sounder (AIRS) launched in May 2002 (Aumann et al, 2003). These missions were followed by dedicated $CO_2$ observers such as the Greenhouse gases Observing SATellite (GOSAT) mission in 2009 (Kuze et

al., 2009) and the Orbiting Carbon Observatory 2 (OCO-2) in 2014. These data have yielded substantial scientific insights, such as a much more dynamic tropical carbon cycle compared with previous understanding (e.g., Liu et al, 2017; Palmer et al, 2019; Crowell et al, 2019; Peiro et al, 2021) as well as studies into power plant emissions and plumes (Nassar et al., 2017).

OCO-2 measures reflected solar radiances, from which column averaged $CO_2$ dry air mole fractions ($XCO_2$) are retrieved with the NASA Atmospheric $CO_2$ Observations from Space (ACOS) algorithm (Crisp et al., 2012; O'Dell et al., 2012; Connor et al., 2008). Radiances are measured in the near-infrared Oxygen A band near 0.76 µm; the shortwave infrared weak $CO_2$ band near 1.6 µm; and the shortwave infrared strong $CO_2$ band near 2.05 µm. ACOS is based on Bayesian optimal estimation (Rodgers, 2000) that adjusts input parameters (e.g., $XCO_2$, aerosols, surface characteristics, surface pressure) to maximize agreement between a modelled spectrum (derived by a radiative transfer model) and OCO-2 measurements. The parameters that best explain the measured radiances are labelled as the "retrieved" parameters. ACOS has undergone continuous improvement since the initial version.

Since the radiances contain uncorrected calibration artifacts and the modelled representation of the atmospheric radiative transfer is not perfect, retrieved parameters contain systematic biases. The inverse problem is under-constrained and leads to posterior errors in retrieved parameters that are correlated. To correct for errors in $XCO_2$ arising from these types of dependencies, a multiple linear regression (MLR) bias correction with co-retrieved state variables or features used as predictors is fit to the difference ($\Delta XCO_2$) between the ACOS retrieved $XCO_2$ and a truth proxy estimate of $XCO_2$. This method was first introduced for ACOS retrievals applied to the GOSAT instrument (Wunch et al., 2011), and later extended to OCO-2 and OCO-3 (O'Dell et al., 2018; Taylor et al., 2020, 2023). The multiple linear regression (MLR) bias correction is fit in tandem with a quality filter of empirically defined thresholds on a set of features. The bias correction and quality filter are derived iteratively, with filter thresholds chosen restricting features to a range in which the relationship between $\Delta XCO_2$ and the parameters are mostly linear, improving the goodness of fit for the multilinear regression, which is then used in turn to retune the quality filter thresholds. The combined bias correction and quality filtering process is derived manually, so that the final product must be hand-tuned for each algorithm update. After the feature-based correction, a footprint correction and global TCCON offset are applied. The combined bias correction and quality filter is robust across a set of ground truth proxy metrics and greatly reduce both mean bias and error scatter of $XCO_2$ retrieved from OCO-2. Full details of the operational bias correction and filtering can be found in O'Dell et al. (2018).

A drawback of applying the quality filter is the exclusion of data due to the linear assumption of the bias correction to which the quality filter limits the regime of interaction between state vector variables and $\Delta XCO_2$, thus removing data where the bias is non-linear. Due to loss of data, the bias correction and quality filter are often disregarded for local studies (Nassar et al., 2017; Mendonca et al., 2021) and are too limiting for certain regions (Jacobs et. al., 2020). Applying non-linear machine learning techniques have shown great promise for the task of bias correction for GOSAT/GOSAT-2 (Noël et al., 2022) and

TROPOMI (Schneising et al., 2019). Specific correction of 3D cloud biases for OCO-2 retrieved $XCO_2$ (Massie et al. 2016) using a non-linear method fit on a small set of features correlated with 3D cloud effects in addition to the linear operational correction, is demonstrated in Mauceri et al. (2023).

This research demonstrates a general non-linear bias correction approach for OCO-2 build 10 (B10, Taylor et al., 2023) via a machine learning method and provides a post-hoc explanation of the overall contribution of the selected state vector features. Our non-linear bias correction is shown to reduce systematic errors and increases the percentage of good quality soundings by allowing for the relaxation of the hand-tuned thresholds employed with the standard quality threshold method. The framework presented in this manuscript for identifying informative features for bias correction can be adapted for future OCO-2,3 ACOS

algorithm updates.

## 2 Data

To develop a bias correction, we define three truth proxy data sets for the true atmospheric column mole fraction. $\Delta XCO_2$ is then set as the difference between the raw ACOS retrieval of $XCO_2$ and the truth proxy estimate of $XCO_2$ as shown in Eq. 1. For the TCCON and model mean truth proxies, the OCO-2 averaging kernel is also applied as described in Taylor et al., 2023.

$$\Delta XCO_2 = XCO_{2,ACOS} - XCO_{2,Proxy} \tag{1}$$

We use the same proxy data sets used in the development of the operational bias correction (Osterman et al. 2020): co-located OCO-2 soundings with Total Carbon Column Observing Network (TCCON), a collection of small-area clusters of soundings for which $XCO_2$ is not expected to vary above the instrument noise, and a set of modelled mole fractions whose underlying surface flux is constrained by the NOAA global in situ network (Masarie, K., et al. 2014). Data sets include soundings from

80 November 2014 through to February 2019. Each truth proxy captures a different scale of retrieval error and as such give complementary information as described in O'Dell et al. (2018). All data sets were sampled in conjunction with corresponding locations and times in the OCO-2 B10 L2 lite files which can be found here: (**https://disc.gsfc.nasa.gov/datasets/**). Spatial coverage and sounding count is shown in Figure 1. The newest version available Level 2 product is build 11 (B11), however at the time of this writing was undergoing re-processing.

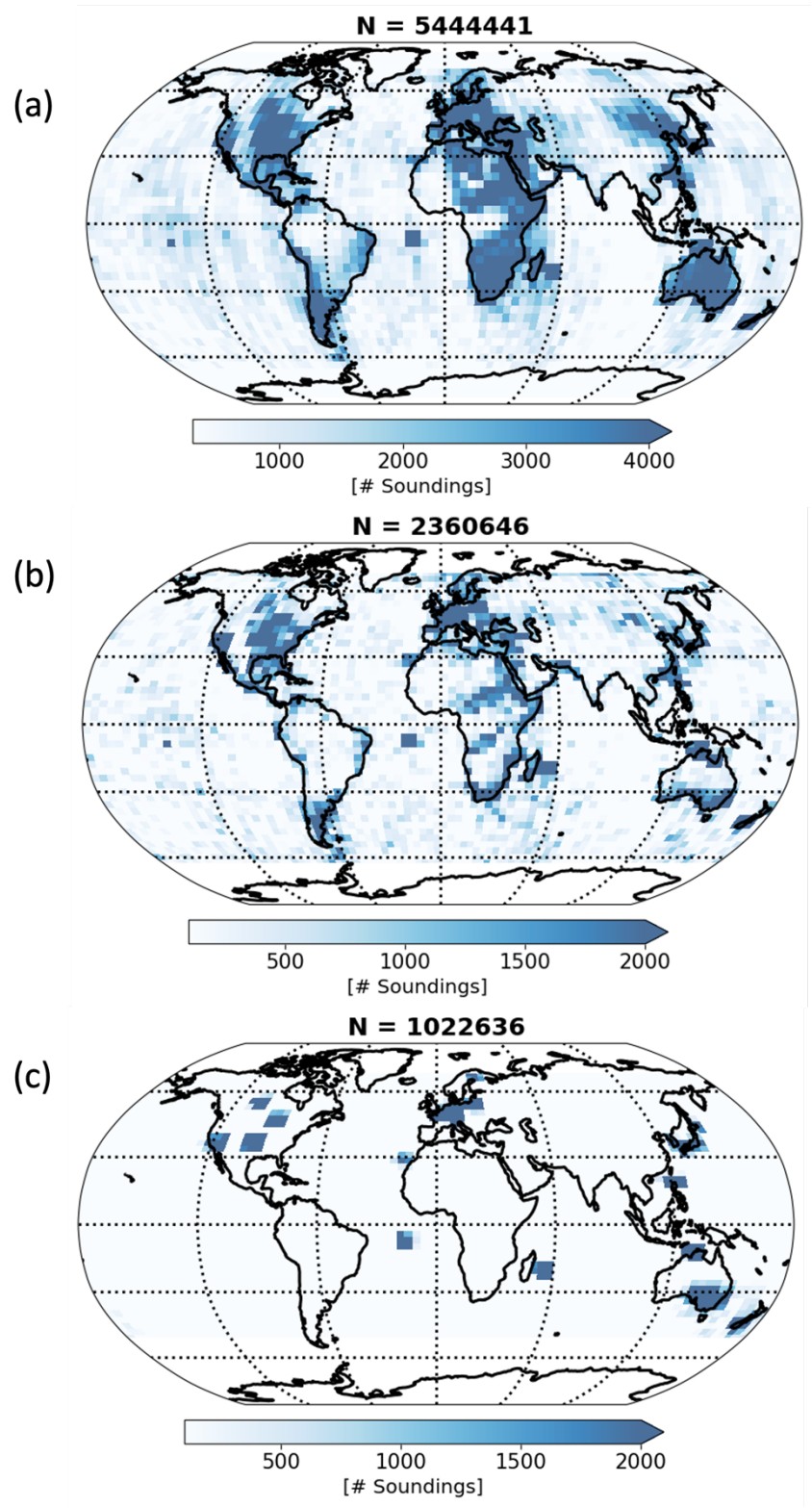

**Figure 1: Spatial coverage for each truth proxy. The mean of a set of flux models is shown in (a), small area approximation is shown in (b), and TCCON is shown in (c).**

## 2.1 TCCON truth proxy

TCCON is a system of ground-based sun-looking Fourier Transform Spectrometers with growing global coverage that retrieve dry air mole column averaged measurements of the trace greenhouse gases from radiances in similar spectral bands to OCO-2. Since each site has been extensively validated against WMO-traceable in situ observations aboard aircraft, TCCON offers the most accurate comparison for $XCO_2$ (Wunch et al., 2010). While TCCON is well calibrated, site coverage is limited outside of North America, Europe, and Oceania. The TCCON data set therefore is spatially the sparsest of the three truth proxies and offers non-uniform point comparisons. We use the same dataset as the operational correction consisting of OCO-2 soundings co-located with TCCON GGG2014 measurements (Wunch et al., 2017; Wunch et al., 2011) in space (2.5º lat, 5º lon) and time (2h).

Table 1. TCCON sites used in bias correction and filtering for B10 ACOS.

| TCCON (station name* = island) | Continent | Latitude | Altitude (m) | Operational date range (YYYYMM-YYYYMM) | Data citation |
|---|---|---|---|---|---|
| Saga* | Asia | 33.2º N | 7 | 201106-present | Shiomi et al. (2014) |
| Orléans | Europe | 48.0º N | 130 | 200908-present | Warneke et al. (2019) |
| Garmisch | Europe | 47.5º N | 740 | 200707-present | Sussman and Rettinger (2018) |
| Tsukuba* | Asia | 36.1º N | 30 | 200812-present | Morino et al. (2018a) |
| Sodankylä | Europe | 67.4º N | 188 | 200901-present | Kivi et al. (2014) |
| Rikubetsu | Asia | 43.5º N | 380 | 201311-present | Morino et al. (2018) |
| Izaña* | Africa | 28.3º N | 2367 | 200705-present | Blumenstock et al. (2017) |
| JPL | N. America | 34.2º N | 390 | 201103-201307 201706-201805 | Wennberg et al. (2017a) |
| Bialystok | Europe | 53.2º N | 180 | 200903-201810 | Deutscher et al. (2019) |
| Bremen | Europe | 53.1º N | 27 | 200407-present | Notholt et al. (2019) |
| Wollongong | Australia | 34.4º S | 30 | 200805-present | Griffith et al. (2014b) |
| Park Falls | N. America | 45.9º N | 440 | 200405-present | Wennberg et al. (2017b) |
| Réunion* | Africa | 20.9º S | 87 | 201109-present | De Maziére et al. (2017) |
| Anmyeondo | Asia | 36.5º N | 30 | 201408-present | Goo et al. (2014) |
| Darwin | Australia | 12.4º S | 30 | 200508-present | Griffith et al. (2014a) |
| Lauder* | Australia | 45.0º S | 370 | 200406-present | Pollard et al. (2019) |

| Lamont | N. America | 36.6º N | 320 | 200807-present | Wennberg et al. (2016) |
|--------|-----------|---------|-----|----------------|------------------------|
| Karlsruhe | Europe | 49.1º N | 116 | 200909-present | Hase et al. (2015) |
| Manaus | S. America | 3.2º S | 49.2 | 201408-201506 | Dubey et al. (2014) |
| Paris | Europe | 48.8º N | 60 | 201409-present | Te et al. (2014) |
| Burgos* | Asia | 18.5º N | 35 | 201703-present | Morino et al. (2018b) |

## 2.2 Small area approximation truth proxy

The small area approximation described in O'Dell et al. (2018) offers insight into small scale drivers of bias and retrieval variability. The small area approximation truth proxy assumes that $XCO_2$ within a 100km neighbourhood is largely uniform for a given overpass by OCO-2. This assumption is evaluated in Worden et al. (2017), where it was found by using a high-resolution atmospheric model (GEOS-5), variance of $XCO_2$ is around 0.1 ppm per 100km. The proxy offers improved spatial coverage compared to TCCON but struggles to capture biases with low variability over the small area.

## 2.3 Flux models truth proxy

A set of flux inversion models form the largest of the truth proxy data sets, both in number of soundings and in spatial coverage. The models included in this truth proxy set are found in Table 2. The posterior $XCO_2$ fields produced by the models are sampled along OCO-2 tracks, the proxy is then computed as the average of the models at every sounding where there is good agreement (within 1.5 ppm) among models (O'Dell et al., 2018; Osterman et al., 2020).

**Table 2. Flux models used for the model mean truth proxy. TM5 – Transport model 5, TM3 – Transport model 3, LMDZ – Laboratoire de Meteorology, EnKF – Ensemble Kalman Filter, 4D-Var – 4 Dimensional Variation.**

| Model name | Institute | Transport model | Resolution [latxlonxtime] | Inverse method | Citation |
|------------|-----------|-----------------|---------------------------|----------------|----------|
| CarbonTracker | NOAA Global Monitoring Laboratory | TM5 | 2ºx3ºx3h | EnKF | Peters et al. (2007) CarbonTracker (2021) |
| CarboScope | Max Planck Institute for Biogeocehmistry | TM3 | 4ºx5ºx6h | 4D-Var | Rödenbeck (2005); Rödenbeck et al. (2018) CarboScope (2021) |
| CAMS | Copernicus Amosphere Monitoring Service | LMDZ | 1.9ºx3.75ºx3h | 4D-Var | Chevallier et al. (2010) CAMS (2021) |

# 3 Methods

## 3.1 Gradient boosting

To model systematic error from co-retrieved state vector elements, we employ a machine learning method known as extreme gradient boosting or XGBoost (Chen et al. 2016) which can fit both linear and non-linear relationships. XGBoost is an ensemble model where a set of simple models known as regression trees (Breiman 1984) are sequentially trained, with each new member fit on residuals of the previous trees. During inference, the weighted sum is taken across the ensemble members. Members are grown or fit by selecting features that provide high information gain (Eq. 2). Information gain is calculated by evaluating the sum of the gradients $G$ and hessians $H$ of the loss function at left and right leaf nodes when selecting split points for a feature during tree fitting. For our experiments we minimize the Mean Squared Error between the truth proxy bias $y_i$ and the estimate $\hat{y}_i$ as the loss function as shown in Eq. 3. Features that are informative for reducing residual error during tree development yield high gain values. These values can be summed across trees in the ensemble to produce a ranking of feature contribution. This provides a post-hoc method of interpretability yielding a high level or global view of feature importance to correcting $\Delta XCO_2$. While this method of interpretability is less informative than the regression coefficients provided by a linear model, it is useful for tasks such as feature selection.

XGBoost employs $L_1$ and $L_2$ norm regularization to reduce overfitting to outliers present in the training dataset. The effect of the regularization is governed by the hyper-parameters $\lambda$ and $\gamma$, and must be carefully selected or tuned. To find these hyper-parameters we use a k-fold cross validation strategy in which the training dataset is divided into $k$ subsets (we use $k=10$) and each subset is sequentially held out for evaluation for a model trained on the rest of the data. Performance across the k-folds is averaged and the process is repeated for each potential selection of hyper-parameters. We found a $\lambda_{LAND}=2.5$ and $\gamma_{LAND}=3.75$ for the land correction, and $\lambda_{OCEAN}=2.0$ and $\gamma_{OCEAN}=10.0$.

$$Information\ Gain = \frac{1}{2}\left[\frac{G_{Left}^2}{H_{Left}+\lambda} + \frac{G_{Right}^2}{H_{Right}+\lambda} - \frac{(G_{Left}+G_{Right})^2}{H_{Left}+H_{Right}+\lambda}\right] - \gamma\ , \tag{2}$$

$$Mean\ Squared\ Error\ loss = \frac{1}{N}\Sigma_{i=1}^{N}(y_i - \hat{y}_i)^2, \tag{3}$$

## 3.2 Quality filtering

Soundings of the lowest quality are typically caught by the $O_2$ A-band pre-processor (Taylor et al. 2012) and IMAP-DOAS (Frankenberg et al. 2005) algorithms due to clouds and low SNR in the continuum and are then screened out before being run through the L2 retrieval algorithm (Taylor et al., 2016). After retrieval, an additional number of soundings are flagged and removed for which the ACOS algorithm failed to converge or for which the chi-squared difference between modelled and measured spectra is too large. Additionally, large unphysical outliers present in the tails of the conditional distributions of several atmospheric state variables are also removed by hand using domain expert selected thresholds. Finally, users can select for high-quality retrievals using the binary $XCO_2$ quality flag (QF) with "good" data having a QF = 0, and "poor" data having QF = 1. The operational $XCO_2$ quality flag is derived using a set of filters applied to the state vector variables found in conjunction with linear parametric bias correction. An initial linear correction is fit on soundings that have passed the pre-processing filtering steps. Each filter is then hand selected, QF = 1 data is removed, and the bias correction is re-fit until a final set of filters and linear model weights are derived that sufficiently reduce mean bias and scatter (O'Dell et al 2018).

To assess the ability of the non-linear method to correct QF = 1 data and the potential for increased throughput of well corrected data, we derive a new quality flag (QFNew). Our flag is developed in a similar fashion to the B10 quality flag for use with the non-linear correction. The first step is to start with the same set of state vector variables and associated thresholds. Next, thresholds are relaxed for a selection of state vector variables that allow for higher sounding throughput, while maintaining or reducing corrected $\Delta XCO_2$ across truth proxies. Thresholds are never set to be more constraining than the B10 values in order to not remove soundings that are already considered to be of passing quality.

## 3.3 Training and test split

For training and evaluating the non-linear correction, we subset each of our truth proxy datasets into training and testing datasets. First, datasets are split by the two surface types: ocean and land. In the B10, both operation modes (nadir and glint) are combined for the land bias correction due to low variance in feature importance between nadir and glint (O'Dell et al. 2018). To compare to the operational correction, we also combine both modes for the land correction model. The land and ocean data sets are subset once more by truth proxy to identify informative features for the final land and ocean models. To ensure that model performance is indicative of how well the models generalize to unseen data, we hold out a year of data for evaluation of the final land model and ocean model. Models are trained on data from 2014, 2015, 2016, and 2017, then evaluated on data from 2018. Since data from 2019 is limited we exclude it from both training and evaluation.

## 3.4 Experiment Design

First, the footprint correction as described in O'Dell et al., 2018 is applied to the training and evaluation datasets. We then evaluate two methods for bias correcting retrieved XCO2: a non-linear machine learning model called XGBoost; and as a baseline, we also train a MLR model similar to the hand-tuned model used in the operational correction. For correcting land nadir, and land glint data, a single XGBoost model and MLR are trained using all three truth proxies. The predictor variables, or features are the same for both model types. This allows for comparison between the non-linear model and baseline linear method to properly assess that improved fit is coming from the captured non-linearity and not just the inclusion of the additional predictors. A single XGBoost and MLR are derived for correcting ocean glint data, again using all three proxies and same set of ocean features. We also compare our approach to the operational land correction and ocean correction for B10.

To identify a set of informative features to be used as inputs for the XGBoost land and ocean models, we first train a set of models independently on each truth proxy. These six models (three for land and three for ocean) are initially fit on a large set of potentially informative features, using QF = 0 + 1 data. The resulting feature importance derived from these initial models is used to filter down the feature set to identify a subset of features that is highly informative across truth proxies. The resulting feature sets are combined to train the final proposed model pair (one for land and one for ocean), which are trained using all truth proxies.

Next, we compare the final models trained on QF = 0 + 1 data against models trained only on "good" quality data assigned QF = 0 then, evaluate each model pair on QF = 0 soundings that have been temporally held out. This is to ensure the ability of the nonlinear method to reproduce the linear model, which is the currently accepted community standard. Secondly, we evaluate the model trained on QF = 0 + 1 data on the excluded regime of data labelled QF = 1 where non-linear relationships between $\Delta$XCO2 and predictors become more pronounced. Finally, we derive a new quality flag (QFNew) used in conjunction with the non-linear correction that increases the throughput of well corrected data while maintaining similar error metrics as the operational filter and correction.

## 4 Results

### 4.1 Feature Selection

We select informative features for our bias correction following an iterative procedure. In the first step, we train XGBoost models for each proxy by surface type and operation mode (6 models in total). These initial models are trained using a large subset of co-retrieved state vector variables (shown in Table C1) which are potentially informative for correcting $\Delta$XCO2 from the B10 L2Lite files. The resulting models are used to rank features according to their information gain which is defined in Eq. 2. Features that are less informative are removed from the set and new models are trained with the reduced feature set. Afterwards, feature importance is once again evaluated. To ensure robustness to correlation among features (which information gain does not account for) we calculate Pearson's correlation values between features. Features with an absolute Pearson value greater than 0.5 are included one a time and the feature with the highest importance is kept. This process is iteratively repeated

until reaching a relatively small subset of maximally informative features. These features are combined to train the final bias correction models, which are trained on all proxies. Seven features are selected for land correction and five features selected for ocean as shown in Figure 2. The resulting features used in the final models and a brief description is shown in Table 3.

220

Features used for the operational correction are also highly informative for the proposed non-linear corrections and include the difference between the retrieved $CO_2$ profile and prior profile used for land and ocean (*co2_grad_del*), and two surface pressure difference terms *dpfrac* for land and *dp_sco2* for ocean (Kiel et al., 2019). The co2_grad_del is the change in profile shape and the prior and is calculated as the difference dry air mole fraction at the surface, denoted as $CO_2(1)$, to the fraction at ~0.6316 times the retrieved surface pressure, and is in units of parts per million (ppm). The calculation for co2_grad_del is shown in Eq. 4. For land, the dpfrac term is a difference ratio that considers the smaller dry air column over higher elevations and is defined in Eq. 5 where $X_{CO2,raw}$ is the uncorrected retrieval of the column average and $P_{ap,SCO2}$ and $P_{ret}$ are the prior surface pressure at the strong band pointing offset and retrieved pressure respectively. For ocean, dp_sco2 is used and is the retrieved surface pressure minus the strong band prior. The extensive use of co2_grad_del and surface pressure deltas for bias correction is discussed in Kulawik et al., 2019.

230

$$co2\_grad\_del = [CO_{2,ret}(1) - CO_{2,ret}(0.6316)] - [CO_{2,prior}(1) - CO_{2,prior}(0.6316)] \qquad (4)$$

$$dpfrac = X_{CO2,raw}(1 - {P_{ap,SCO2}}/{P_{ret}}) \qquad (5)$$

235

For land, the *h2o_ratio* is used and is the ratio of $XH_2O$ estimated by single band retrievals from the strong and weak $CO_2$ bands separately using the IMAP-DOAS algorithm, which can differ from unity in the presence of atmospheric scattering (Taylor et al. 2016). We use three aerosol features for our bias correction over land scenes. The first being the sum of dust, water, and sea salt optical thickness termed *DWS*. We include retrieved ice particle optical depth (*aod_ice*) and the finer stratospheric aerosol optical depth (*aod_strataer*). The last feature used for land, as well as for ocean, is the albedo slope for the strong $CO_2$ band termed *albedo_slope_sco2*. This variable represents the slope of the reflectance across the strong $CO_2$ spectral band for land soundings and the slope of the Lambertian component of the combined Cox-Munk and Lambertian Bidirectional Reflectance Distribution Function (BRDF) for ocean soundings (Cox and Munk, 1954). In addition to the co2_grad_del, albedo_slope_sco2 and dp_sco2, two additional variables are used for the correction of Ocean G scenes. These are *snr_wco2*, which is the estimated signal to noise ratio derived during optimal estimation; and finally, *rms_rel_wco2* which is the percent residual error from the forward modelled radiance for the weak $CO_2$ to the measured radiance.

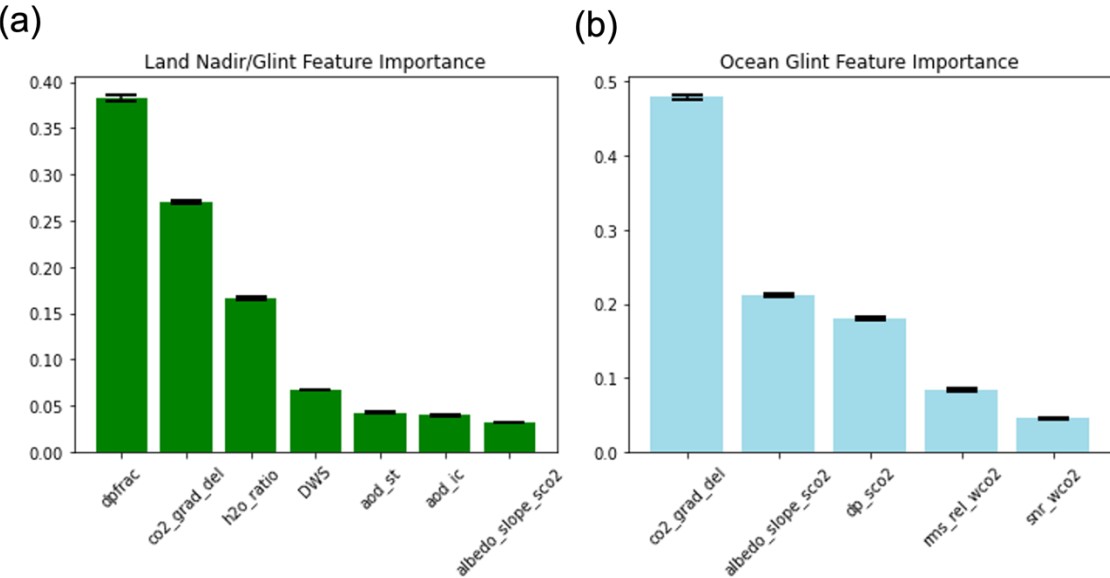

**Figure 2. Feature importance for final land model trained with all proxies (a), feature importance for final ocean model trained with all proxies (b). Error bars denote variance in feature importance across 10 runs with different random seeds.**

**Table 3. Selected features for use in our bias correction models. The first column shows state vector variable names as defined in the B10 L2 files, second provides a brief description, and the last column shows which region and viewing mode correction the variable is used for.**

| State Variable | Description | Surface Type & Operation Mode |
|---|---|---|
| **dpfrac** | Surface pressure difference that considers smaller dry air columns over higher elevations (Kiel et al. 2019). | Land NG |
| **h2o_ratio** | Ratio of retrieved $H_2O$ column in weak and strong $CO_2$ bands by IMAP-DOAS. | Land NG |
| **DWS** | Additive combination of retrieved dust, water, and sea salt aerosol optical depth. | Land NG |
| **aod_strataer (aod_st)** | Retrieved upper tropo+stratospheric aerosol optical depth at 0.755 microns. | Land NG |
| **aod_ice (aod_ic)** | Retrieved ice cloud optical depth at 0.755 microns. | Land NG |
| **co2_grad_del** | Difference between the retrieved vertical $CO_2$ profile and prior. | Land NG/Ocean G |
| **albedo_slope_sco2** | Retrieved strong band reflectance slope(land) or slope of Lambertian albedo component of BRDF (ocean). | Land NG/Ocean G |

| dp_sco2 | Surface pressure difference between the retrieved and prior, evaluated for the strong $CO_2$ band location on the ground. | Ocean G |
|---|---|---|
| rms_rel_wco2 | RMSE of the L2 fit residuals in the weak $CO_2$ band relative to the signal. | Ocean G |
| snr_wco2 | The estimated signal-to-noise ratio in the continuum of the weak $CO_2$ band. | Ocean G |

260

## 4.2 Model evaluation for QF = 0

To ensure that the non-linear method generalizes the linear relationships largely observed for QF = 0, we evaluate two XGBoost
models: one which is fit on QF = 0 + 1, and one fit on QF = 0, to a MLR fit on the same feature set as the non-linear models.
As the operational quality flag is hand-tuned by re-fitting a MLR, the regime between the variables selected for correction and
systematic error are reduced to mostly linear relationships. The non-linear method has only a marginal improvement over the
MLR and B10 correction on soundings that are passed by the operational quality filter over land (**0.02-0.04 ppm**), and a slightly
more substantial improvement over ocean (**0.09-0.10 ppm**), on the evaluation data. We found that retraining the XGBoost
models on QF = 0 data does not offer a substantial reduction in error despite initial XGBoost models being trained on un-
filtered data. We forgo the iterative refitting approach that is required for the MLR and operational correction by training once
on QF = 0 + 1 data. Table 4 shows the QF = 0 RMSE results for XGBoost models trained on both QF = 0 + 1 data and QF =
0 data, alongside the MLR model fit to the filtered regime for 2018 and B10 operational correction.

**Table 4: RMSE scores for 2018 on QF=0 data. Results are shown for Land and Ocean data by truth proxy and model. Two XGBoost
models are shown: one trained on QF = 0 + 1 (XGBoost$_{QF0+1}$) data and the evaluated on QF = 0, and another (XGBoost$_{QF=0}$) trained
and evaluated on only QF = 0 data. A multiple linear regression (MLR$_{QF=0}$) is also fit for QF = 0 using the same feature set. In the
last column, RMSE for operationally corrected XCO$_2$ (B10) is shown.**

| | Land QF=0 | | | |
|---|---|---|---|---|
| **Truth Proxy** | **XGBoost$_{QF=0+1}$ RMSE** | **XGBoost$_{QF=0}$ RMSE** | **MLR$_{QF=0}$ RMSE** | **B10 RMSE** |
| Small area | 0.83 ppm | 0.82 ppm | 0.84 ppm | 0.85 ppm |
| TCCON | 1.15 ppm | 1.14 ppm | 1.19 ppm | 1.20 ppm |
| Model mean | 1.05 ppm | 1.05 ppm | 1.09 ppm | 1.11 ppm |
| All | 1.03 ppm | 1.02 ppm | 1.05 ppm | 1.07 ppm |

| Ocean QF=0 | | | | |
|---|---|---|---|---|
| **Truth Proxy** | **XGBoost$_{QF=0+1}$ RMSE** | **XGBoost$_{QF=0}$ RMSE** | **MLR$_{QF=0}$ RMSE** | **B10 RMSE** |
| Small area | 0.45 ppm | 0.44 ppm | 0.56 ppm | 0.52 ppm |
| TCCON | 0.83 ppm | 0.81 ppm | 0.89 ppm | 0.95 ppm |
| Model mean | 0.67 ppm | 0.66 ppm | 0.78 ppm | 0.76 ppm |
| All | 0.65 ppm | 0.65 ppm | 0.75 ppm | 0.74 ppm |


### 4.3 Correcting Outside of the Filtered Regime

Correction of systematic error outside of the quality filtered regime (QF = 1) is difficult to fit with a linear model. Strong non-linearities are observed for many of the co-retrieved state vector variables and $\Delta XCO_2$. For many variables this behaviour is

observed over un-physical values in a few spurious soundings and are easily filtered out. Variables such as h2o_ratio which are responsible for the bulk of the quality filtering (h2o_ratio thresholds remove ~10% of soundings) exhibit such non-linear characteristics over their marginal distributions. The dependent linear correction and quality filter is prohibitive for correcting and passing data in these regions of the domain. Figures 3 and 4 illustrate the interaction between state variables chosen for correction and $\Delta XCO_2$. The non-linear model (green) improves both mean and variance of $\Delta XCO_2$ over both the raw $\Delta XCO_2$

(red) before correction, and B10 correction (blue). Table 5 displays the RMSE scores of the XGBoost corrected $XCO_2$ and operational corrected $XCO_2$ for QF = 1 data. The non-linear correction provides a large improvement in reducing the residual error for QF = 1 data over the operational correction with a **1.33-2.26 ppm** improvement for land data and **1.11-1.38 ppm** for ocean. These errors are still significantly larger than the corresponding QF = 0 errors.

**Table 5: RMSE scores for 2018 on QF = 1 data. XGBoost corrected XCO$_2$ and operationally corrected XCO$_2$ (B10) for Land and Ocean data.**

| Land QF = 1 | | |
|---|---|---|
| **Truth Proxy** | **XGBoost$_{QF=0+1}$ RMSE** | **B10 RMSE** |
| Small area | 1.92 ppm | 3.25 ppm |
| TCCON | 2.81 ppm | 5.07 ppm |
| Model mean | 2.46 ppm | 3.95 ppm |
| **Ocean QF = 1** | | |
| **Truth Proxy** | **XGBoost$_{QF=0+1}$ RMSE** | **B10 RMSE** |
| Small area | 1.25 ppm | 2.36 ppm |
| TCCON | 1.68 ppm | 2.90 ppm |

| Model mean | 1.53 ppm | 2.91 ppm |
|---|---|---|

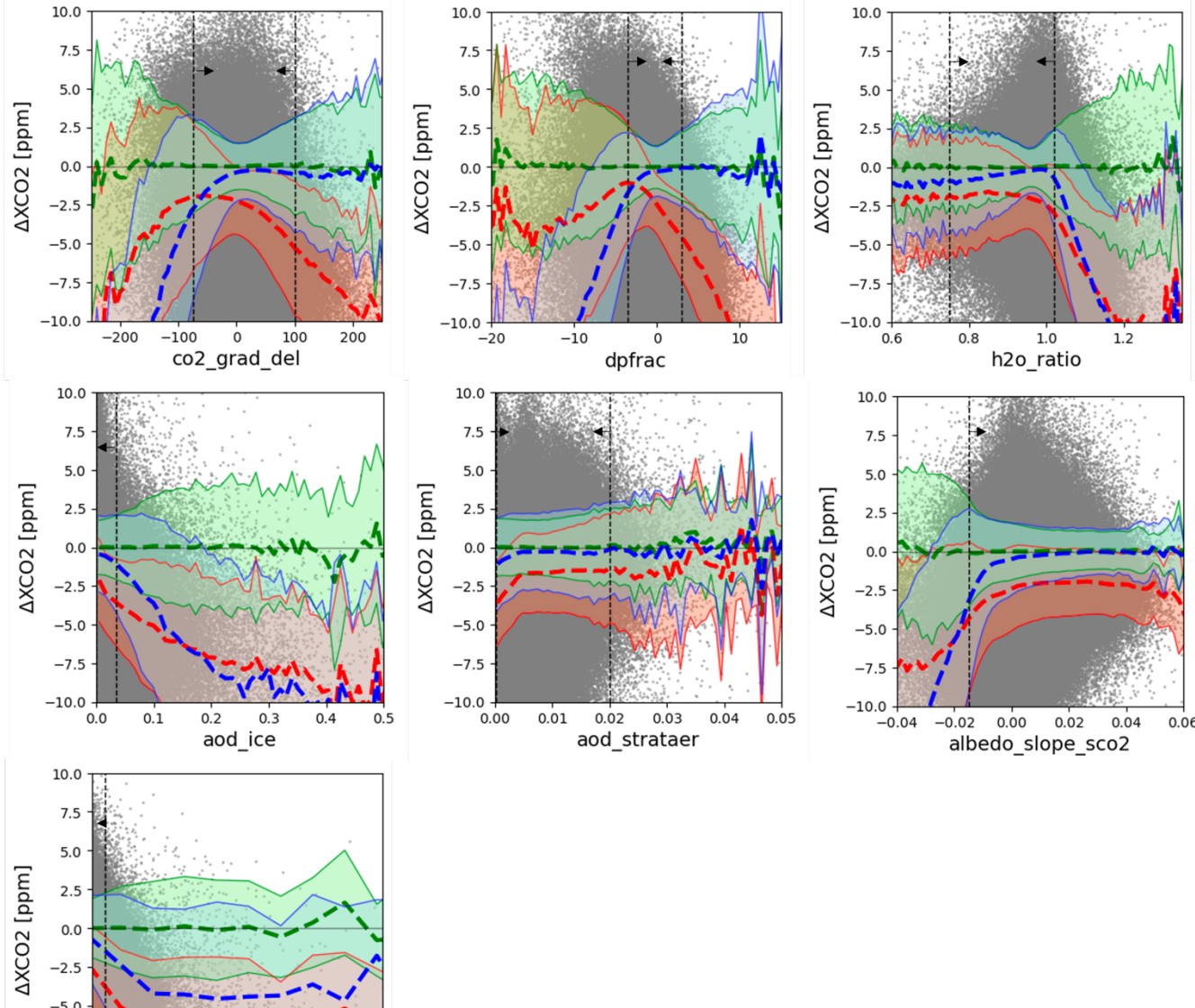

**Figure 3. ΔXCO₂ vs land features for 2018. Mean interaction and 2σ Stddev for uncorrected ΔXCO₂ plotted in red, XGBoost corrected in green and B10 corrected in blue. The vertical black dotted lines indicate B10 QF filters and arrows point towards the region assigned QF=0. Individual soundings are shown with grey scatter.**

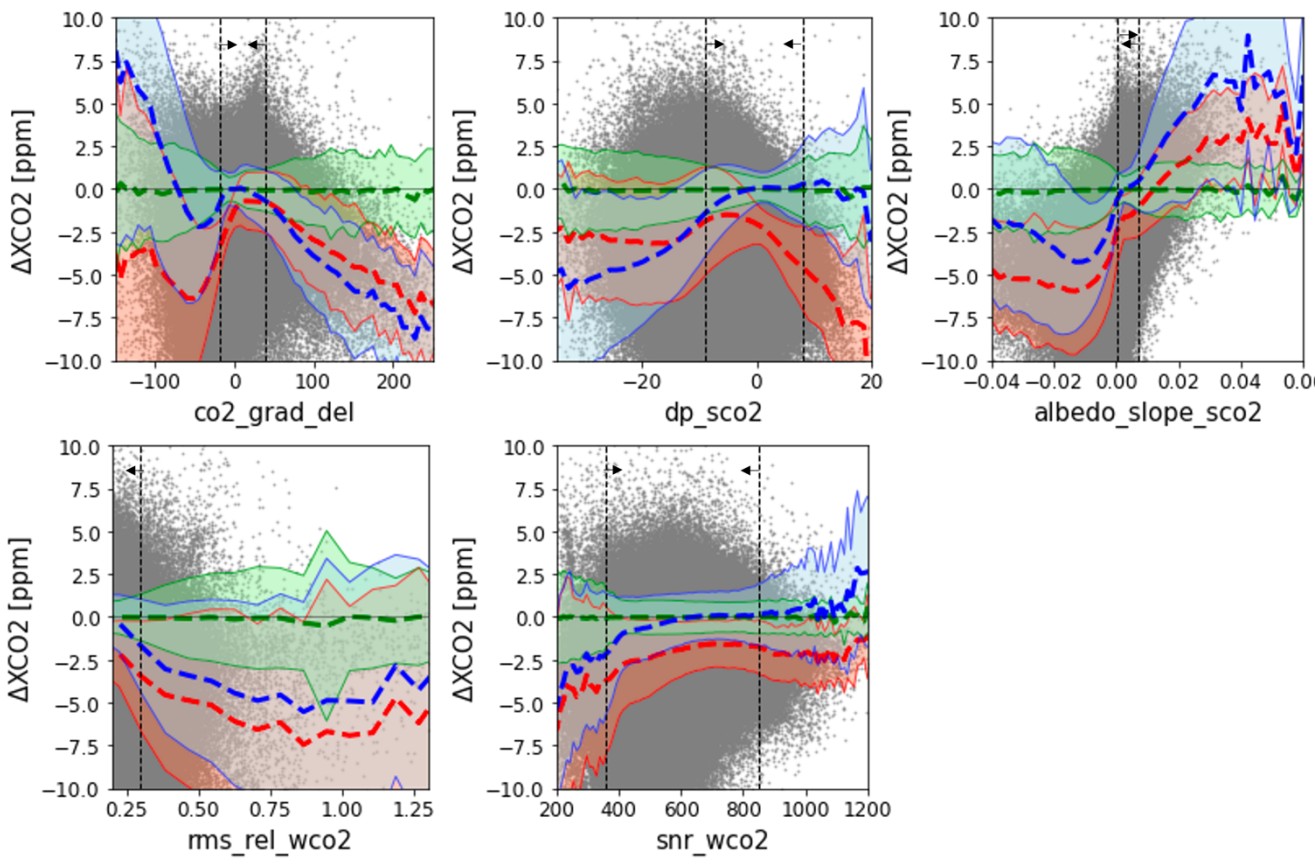


**Figure 4. ΔXCO₂ vs ocean features for 2018. Mean interaction and 2σ Stddev for uncorrected ΔXCO₂ plotted in red, XGBoost corrected in green and B10 corrected in blue. The vertical black dotted lines indicate B10 QF filters and arrows point towards the region assigned QF=0. Individual soundings are shown with grey scatter.**


### 4.4 Comparison to B10

For the operational correction, regression weights for the linear model are hand-selected that have good agreement in their correction across truth proxies. The full operational correction also includes a fixed correction for each of OCO-2's eight footprints as described in Osterman et al. 2020. To provide a fair comparison between the full correction models, we also apply

the footprint correction after applying the non-linear feature correction. Table 6 shows the mean and 1σ standard deviation for each bias correction and QF regime. The largest improvement in the non-linear method over B10 comes when correcting QF=1 data. Achieving a 59% improvement in reduction of error variance for land, and a 67% improvement for ocean data

respectively. The improvement in correction over B10 is less significant for QF=0 with improvement of 8% for land and 19% for ocean.


Regionally, the non-linear correction shows up to a 0.5 ppm improvement over northern Africa, where the B10 correction appears to underestimate $\Delta XCO_2$ in comparison. A reduction in biases is also observed in large parts of South America's tropical and sub-tropical regions as well as parts of tropical Asia shown in Figure 5a. These regions also contain the largest difference in Land NG correction between the methods with an average difference (B10-XGBoost) of -0.5 ppm. There is a slight positive difference between methods over the Amazon Basin and Congo Rainforest (Figure 5e). Figures 5c, and 5d illustrate the improvement of the non-linear method to correct QF=1 data over the operational approach. For QF = 1, where the interaction between features and error is non-linear, large biases in $XCO_2$ remain after operational correction. The XGBoost model reduces these remaining biases in many regions, indicating that there may still be usable data that is filtered out by the operational QF when paired with the non-linear correction.


**Table 6. Comparison of combined proxy mean and standard deviation XGBoost corrected $XCO_2$, $XCO_2$ after the operational correction (B10) and un-corrected $XCO_2$ (Raw) for 2018 and all QF filter regimes for both Land and Ocean data.**

| QF = 0 | | | |
|---|---|---|---|
| **Surface/Mode** | **XGBoost** | **B10** | **Raw** |
| Land NG | -0.04±1.02 ppm | -0.13±1.06 ppm | -1.90±1.68 ppm |
| Ocean G | 0.02±0.64 ppm | 0.18±0.71 ppm | -1.61±1.10 ppm |
| **QF = 1** | | | |
| **Surface/Mode** | **XGBoost** | **B10** | **Raw** |
| Land NG | 0.01±2.45 ppm | -1.24±3.83 ppm | -2.83±3.69 ppm |
| Ocean G | -0.06±1.50 ppm | -1.17±2.59 ppm | -2.79±2.75 ppm |
| **QF = 0 + 1** | | | |
| **Surface/Mode** | **XGBoost** | **B10** | **Raw** |
| Land NG | -0.03±1.75 ppm | -0.59±2.64 ppm | -2.78±2.73 ppm |
| Ocean G | -0.01±1.07 ppm | -0.36±1.85 ppm | -2.09±2.03 ppm |

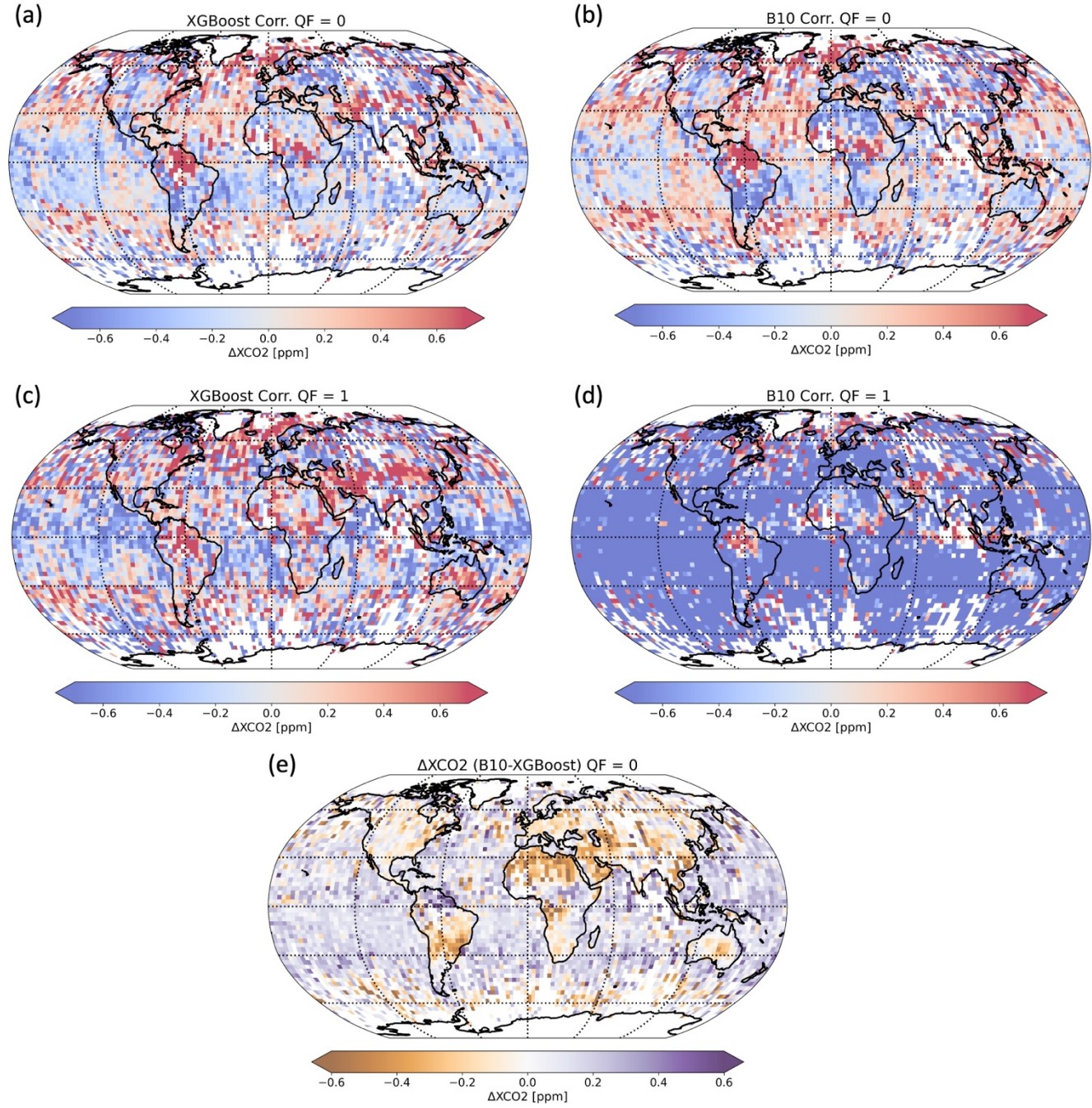

**Figure 5. Remaining XCO₂ biases (ΔXCO₂) after correction for 2018 and model mean proxy, binned to a 3°x3° resolution. ΔXCO₂ after the XGBoost correction for QF=0 is shown in (a), ΔXCO₂ after the B10 correction for QF=0 is shown in (b), ΔXCO₂ after the XGBoost correction for QF=1 is shown in (c), ΔXCO₂ after the B10 correction for QF=1 is shown in (d), and difference (B10 – XGB) for QF=0 is shown in (e).**


## 4.5 Increased sounding throughput

One of the benefits of the non-linear bias correction is the potential for increased throughput of well-corrected QF = 1 data. Improved throughput of well corrected data would be of benefit to point analysis studies where data is limited by the operational QF, and potentially of benefit to flux models as well. To provide an empirical example of this, we create a modified version of the operational $XCO_2$ quality flag utilizing our proposed ocean correction model and land correction model. We take a conservative approach where initial filter values are set equal to those of the operational quality filtering. Then, we select a few variables for which the filters are relaxed to increase sounding throughput while maintaining the RMSE of the combined operational correction and quality filter. With our new quality flag (QFNew), we are able to increase sounding throughput by approximately **14%** over the B10 QF while matching the RMSE of the B10 correction as shown in Table 7.

For many features, the quality filters were not changed from the operational filters, as relaxing filters on variables that are already passing most of their conditional distributions would allow for only marginal improvements in throughput at the cost of large systematic errors. Therefore, we select only features for which large portions of the marginal distributions are removed by the operational flag and where the non-linear correction improves both mean and variance of $\Delta XCO_2$. The relaxed filters for these variables are shown in Figure B1 and Figure B2 by the vertical red dashed lines, and the range of data assigned QFNew = 0 shown in the red parentheses. The operational filter also minimizes the unit-less metric of the binned standard deviation of $\Delta XCO_2$ divided by the posterior $XCO_2$ uncertainty below a value of 3 ppm/ppm (Osterman et al. 2020). When tuning QFNew, we also aim to minimize this metric. Higher throughput of well-corrected data is observed in northern and central Africa, the Amazon basin, and in latitudes above 60º north as seen in Figure 6. While selection of these variables and the relaxation of their filter values is subjective, this empirical result illustrates the benefit of a quality flag derived in conjunction with the non-linear bias correction. Future work will focus on the automation of defining the quality flag thresholds using a data driven approach.

**Table 7. RMSE for combined XGBoost correction, B10 QF percent data throughput, and QFNew percent data throughput, by surface/mode, for 2018.**

| Surface (Mode) | XGBoost RMSE | B10 % Passing | QFNew % Passing |
|---|---|---|---|
| Land (Nadir+Glint) | 1.07 ppm | 59% | 69% |
| Ocean (Glint) | 0.72 ppm | 60% | 74% |

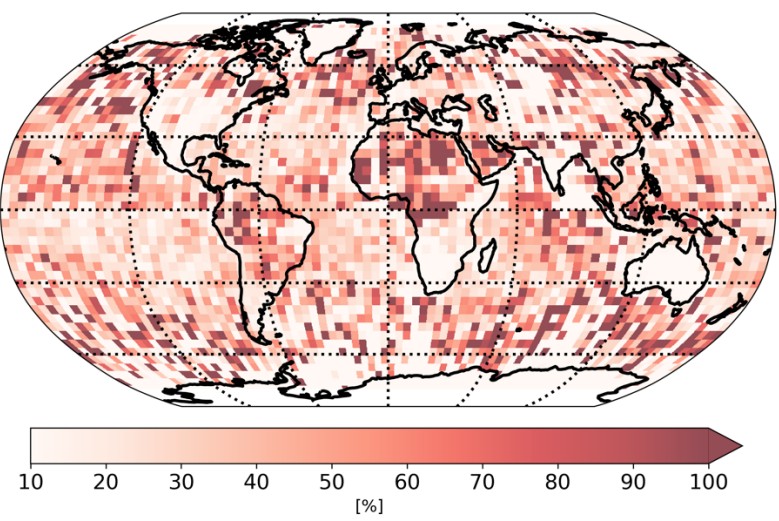

 **Figure 6. Relative increase in percent passing QFNew over B10 QF for 2018 aggregated by 4º×4º bins.**

## 5 Discussion and Future Work

### 5.1 Generalization across proxies

We acknowledge that even with a temporal training and testing split, there is still some circularity due to the lack of a truly independent truth proxy. This issue has been discussed at length for the operational bias correction in Taylor et al. 2023 and comparison and selection of independent validation data sets is still an open area of study. The risk of overfitting due to circularity becomes greater when fitting a more complex machine learning model. To evaluate generalizability to a fully

independent validation proxy, we fit a set of XGBoost models on two truth proxies and evaluate on the third proxy which is held out during training. The same temporal split is used where 2018 data for the held-out proxy is used for evaluation. Results are shown in Figure 7, for land, and Figure 8, for ocean. Each column shows the residual fit for the hold out proxy, for QF = 0 (top row) and QF = 1 (bottom row). For QF = 0, increase in RMSE was minimal for both surface types and across proxies. There was some impact to performance on QF=1 data, when compared to training with all three proxies, particularly for

TCCON with an increase in RMSE of ~0.1 ppm for land and ocean data. Indicating that the information contained in TCCON

is not adequately represented by the model mean and small area approximation proxies which capture variability at larger scales. A potential approach to reducing circularity in the evaluation of the truth proxies would be to train the bias correction on TCCON and either the model mean or small area approximation, using the third proxy not chosen for validation.

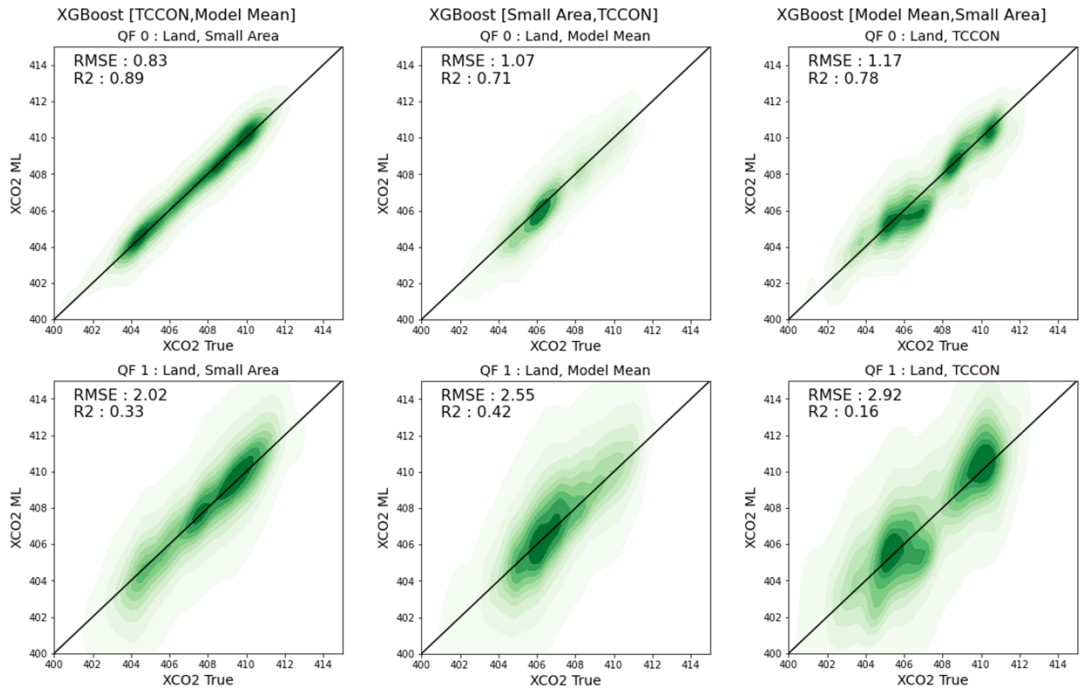

Figure 7. Comparison of XCO$_2$ derived from XCO$_2$ corrected by XGBoost (XCO$_2$ ML) vs. truth proxy (XCO$_2$ True) for land by hold out proxy set and hold out year (2018). Left-most column displays results of a XGBoost model trained on [TCCON, Model Mean] and evaluated on Small Area. Middle column displays results of a XGBoost model trained on [Small Area,TCCON] and evaluated on Model Mean. Right-most column displays results of a XGBoost model trained on [Model Mean, Small Area] and evaluated on TCCON. Generalization for the hold proxy and QF=0 is shown in the top row and QF=1 in the bottom.

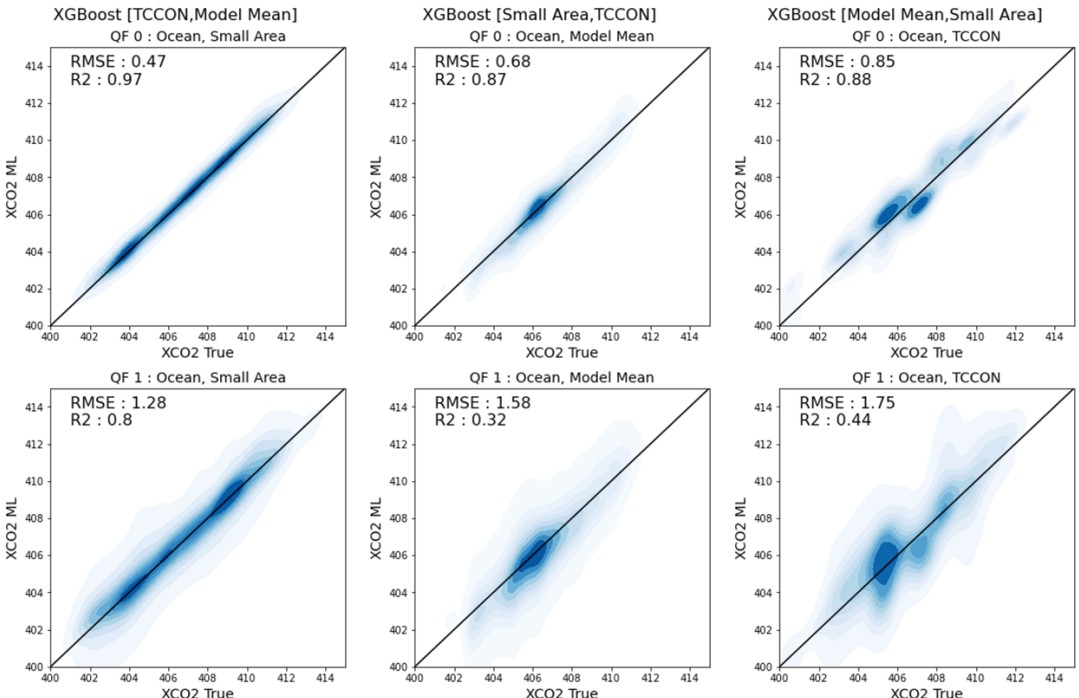

**Figure 8. Comparison of XCO$_2$ derived from XCO$_2$ corrected by XGBoost (XCO$_2$ ML) vs. truth proxy (XCO$_2$ True) for ocean by hold out proxy set and hold out year (2018). Left-most column displays results of a XGBoost model trained on [TCCON,Model Mean] and evaluated on Small Area. Middle column displays results of a XGBoost model trained on [Small Area,TCCON] and evaluated on Model Mean. Right-most column displays results of a XGBoost model trained on [Model Mean, Small Area] and evaluated on TCCON. Generalization for the hold proxy and QF=0 is shown in the top row and QF=1 in the bottom.**

## 5.2 Evaluating feature importance between filter regimes

To understand the contribution of the features to correcting bias in QF=0 and QF=1 data, we compare the information gain between the two regimes. To perform the ablation study, we again employ the models trained on individual truth proxies and re-train and evaluate them on QF=0 and again for QF=1 data. Figure 9 shows the information gain for each filter regime for land and for ocean. For land, dpfrac and co2_grad_del are highly informative for correction of QF=0 data by the machine learning model. Similarly for ocean QF=0 data, the surface pressure delta term dp_sco2 and co2_grad_del are also highly informative. In operation, these terms are also used for bias correction in all ACOS versions (dpfrac replaced dP in B9) to date. These variables are responsible for the largest reduction in unexplained variance in the filtered regime (Payne et al. 2022; Osterman et al. 2020; O'Dell et al. 2018)

For land QF=1 data, there is a drop in importance for co2_grad_del and dpfrac and large increase for h2o_ratio and relative increases for the albedo and aerosol terms. To explain the high importance for the h2o_ratio, we look to the non-linear interaction outside of the bound imposed by the operational filter which removes soundings with a h2o_ratio greater than 1.023, reducing the regime of interaction to one that is not highly correlated with $\Delta XCO_2$. In the QF=1 regime, h2o_ratio corresponds to a significant negative bias. Larger values of h2o_ratio are explained in Taylor et al. 2016, where it was shown

that retrieved surface albedo from the strong $CO_2$ band is generally lower than the weak $CO_2$ band. In cases of larger aerosol presence, this sensitivity leads to weaking of the absorption features and a positive departure from unity. The additional albedo term for the strong $CO_2$ band as well as the additional aerosol terms also increase in importance for QF=1.

    For ocean QF=1 data, there is a significant change in information gain for several features. The surface pressure delta term

dp_sco2, becomes significantly less informative for correcting QF=1 where negative values of dp_sco2 are relatively uncorrelated with $\Delta XCO_2$. Similarly, to land, the albedo term for the strong $CO_2$ band more informative for correcting outside the filtered regime along with the residual error between forward modelled radiances and measurements in the weak $CO_2$ band.

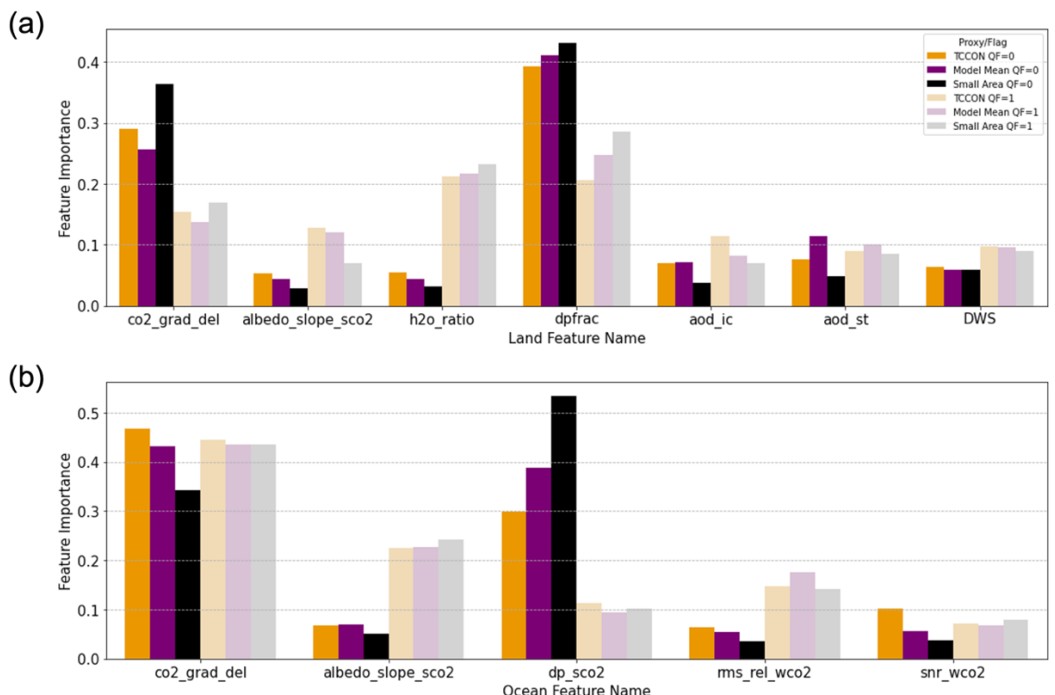

**Figure 9: Feature importance for land is shown in (a), feature importance for ocean is shown in (b). Y-axis displays the normalized information gain from XGBoost models with QF=0 shown in darker colours and QF=1 shown in lighter colours.**

## 5.3 Preservation of CO₂ enhancements

We assess the risk of the proposed bias correction to correct out and remove plume features in the data. Several features heavily utilized by the XGBoost models and in operational correction such as the $CO_2$ gradient delta, and surface pressure terms (e.g., dpfrac, dp_o2a), are differences between the ACOS retrieved state, and the prior. Therefore, there is potentially a risk for the bias correction to use the delta terms to over correct the retrieved $XCO_2$ to the truth. We compare XGBoost corrected $XCO_2$ for two known plumes first identified in Nassar et al. 2021. The two example plumes are shown in Figure 10 (a) and (b): an

ocean glint and land nadir plume in Taean, South Korea, and a land nadir plume observed over two co-located power plants in Ohio, US. We compare the uncorrected $XCO_2$ retrieval (B10 Raw), the operationally corrected $XCO_2$ (B10 Corrected) and the machine learning corrected $XCO_2$ (XGBoost Corrected) and note that the machine learning corrected product captures enhancements not present in the training data. These results are also consistent with the findings in Mauceri et al. 2023 which include similar delta terms. This is further illustrated with the Taean plume which consists of ~35% QF = 0 soundings and ~65

QF = 1 soundings. QFNew = 0 improves the passing rate to ~ 60% as shown in Figure 10 (c). The red stars show data that is passed by QF = 0 (and by construction QFNew = 0) and the blue stars show data that would be removed by QF = 1 but is passed by QFNew = 0, indicating where the increase of available data for the plume feature. Of particular interest is the increase of data within the feature around 36.95° which includes maximum observed enhancement value.

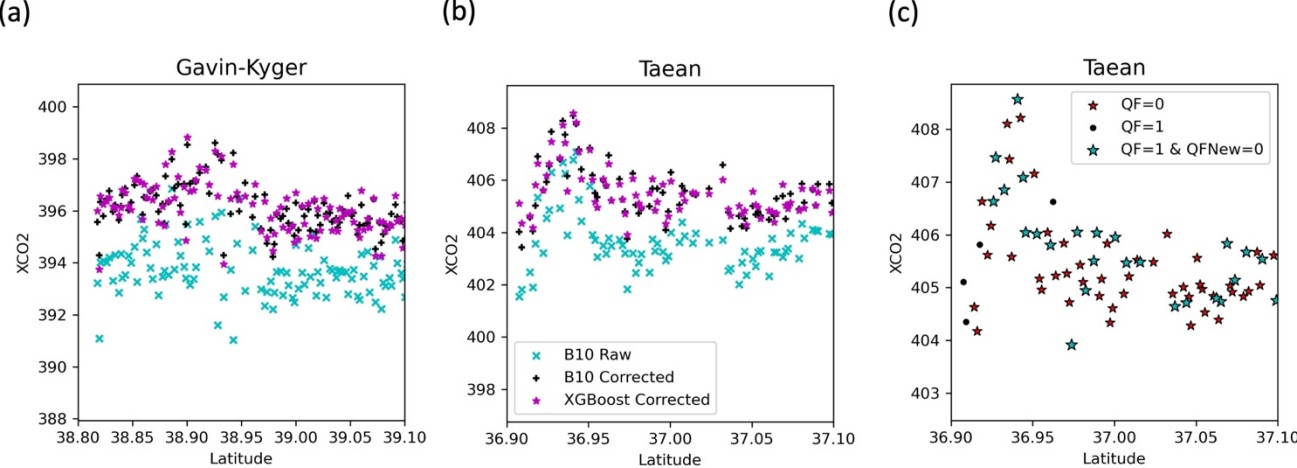

**Figure 10. Two CO₂ plumes captured downwind from power plants (Nassar et al. 2021). An ocean glint and land nadir plume at Taean, South Korea, [lat 36.91°, lon 126.23°] on 2015-04-17 is shown in (a). A land nadir plume near the J. M. Gavin and Kyger Creek power plants in Ohio, USA, [lat 38.93°, lon -82.12°] on 2015-07-30. Regions with the example plumes are not present in the training dataset and consist of QF = 0 + 1 data. Plot (c) shows the increase in XGBoost corrected data for QFNew=0 that would be filtered by the B10 QF.**


## 5.4 Potential for further improving data throughput

Figure 11 further illustrates how the shape of the filtering or decision surface can affect data throughput. Soundings are binned by two state vector features: h2o_ratio and dpfrac. Figure 11b, and Figure 11d show the improvement in reduction of mean $\Delta XCO_2$ and in the error divided by the posterior uncertainty, from the non-linear correction. The QF filters for each feature are indicated by the black dashed lines and the interior of the intersection of these filters indicates the region of state space that is labelled as QF = 0 (Note: the additional filters of the QF further reduce the data that is passed in this region). Significant portions of the distribution, where the non-linear method can accurately correct, lay outside of this filtered region and are labelled QF = 1. A data driven filter can be constructed using similar interpretable machine learning techniques and produce a unified correction/filtering product. Furthermore, moving away from the binary quality flag to a ternary ("very good", "good", "bad") will likely provide an improved data product for end users. Data driven methods for quality filtering have already proven to be useful in the northern high latitudes (Mendonca et al. 2021) and a genetic algorithm was previously used to derive the Warn Levels which complement the operational quality flag found in early OCO-2 data versions (Mandrake et al. 2015). An important task for such future work will be to ensure that the machine learning method learns a physically consistent filter that can increase data throughput while still limiting variance of error and $\Delta XCO_2$. We also acknowledge that while the Taean plume shown in Figure 10 illustrates an empirical example of the ability of a non-linear correction to improve throughput of good quality data, further evaluation of the intersection (QF = 1 & QFNew = 0) will be required before bringing such a method to operation.

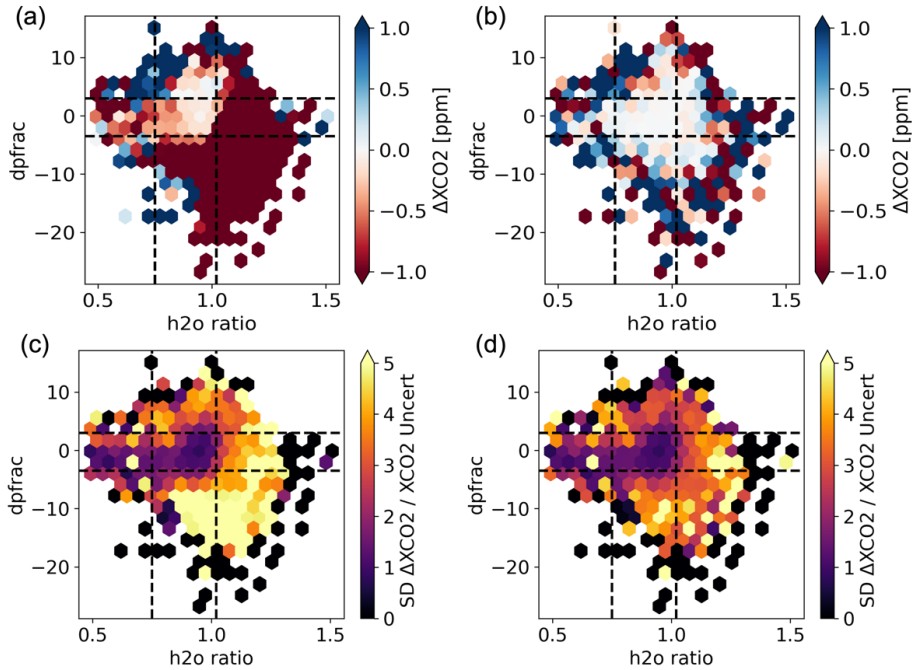


**Figure 11. Hex bin plots show conditional distributions of 2018 $\Delta XCO_2$ vs. dpfrac and h2o_ratio. Remaining $\Delta XCO_2$ after the operational correction for B10 is shown in (a). Remaining $\Delta XCO_2$ after the non-linear correction is shown in (b). Binned sttdev of $\Delta XCO_2$ divided by the posteriori uncertainty from the retrieved $X_{CO2}$ is shown in (c) for the operational correction for B10 and (d) for the non-linear correction. B10 QF filter thresholds for both features are shown with black dashed lines for reference.**


## 6 Conclusion

We demonstrate an approach for selecting co-retrieved state vector variables and other features to be used as input into a land
model and an ocean model to correct biases in ACOS retrieved $XCO_2$. The use of the non-linear method allows for decoupling of the dependent bias correction and filter used in operation, as the filter no longer needs to limit the correction function to a linear fit. By doing so, this method achieves a 59% and 67% improvement in reduction of the error variance over the operational correction on QF=1 data, for land and ocean respectively. To utilize this improvement in correction, we derive a new quality flag (QFNew), by relaxing select filter thresholds from the operational quality flag. Using the proposed QFNew flag, we
increase data throughput by 14% while maintaining a comparable residual error to the operational B10 correction. The workflow outlined in this research is extendable for future ACOS algorithm updates, and for OCO-2's companion instrument, OCO-3, aboard the International Space Station.

## Appendix A: Feature selection and importance


To assess the robustness of our choice of features, we compare the ranking produced by the information gain feature importance generated by the gradient booster, with the ranking produced by a method called permutation feature importance (Fisher et al. 2018). Permutation feature importance captures the contribution to residual error when a feature has its values randomly shifted across observations. Permutation feature importance is a model agnostic post-hoc method that does not require the bias

correction model to be retrained. In Figure A1 we compare the normalized rankings for the individual proxy/surface/mode models that were used to select variables for the final bias correction models trained on all truth proxies. Good agreement is observed in both the overall ranking and magnitude of normalized feature importance between both methods.

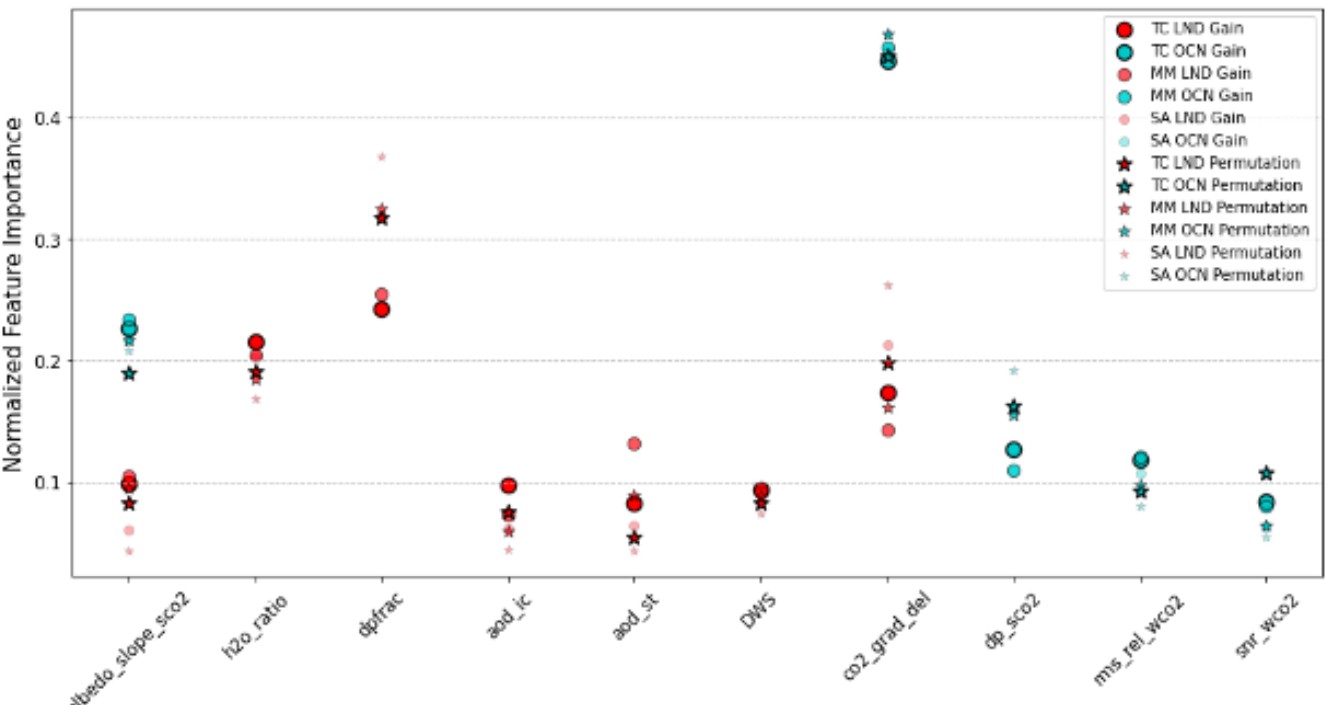

**Figure A1. Comparison of feature importance derived from information gain and permutation importance. Normalized importance (permutation importance in stars, and information gain in circles) are shown for land and ocean features, and by truth proxy. The feature importance produced by both methods are largely in agreement in ranking and overall contribution.**

Feature importance for models trained on individual proxies and QF = 0 + 1 data. These models were used to identify state

variables to be used as input into the proposed bias correction models. While there is generally good agreement between the proxies the overall magnitude and ranking differs slightly as shown in Figure A2. For TCCON the aerosols and albedo terms contribute more to the correction while the same terms are less informative for the small area approximation, which is likely

due to the small area proxy capturing biases that vary slowly over larger scales. For ocean, the albedo_slope_sco2 is informative for the small area proxy, and all proxies exhibit better agreement in their feature importance.


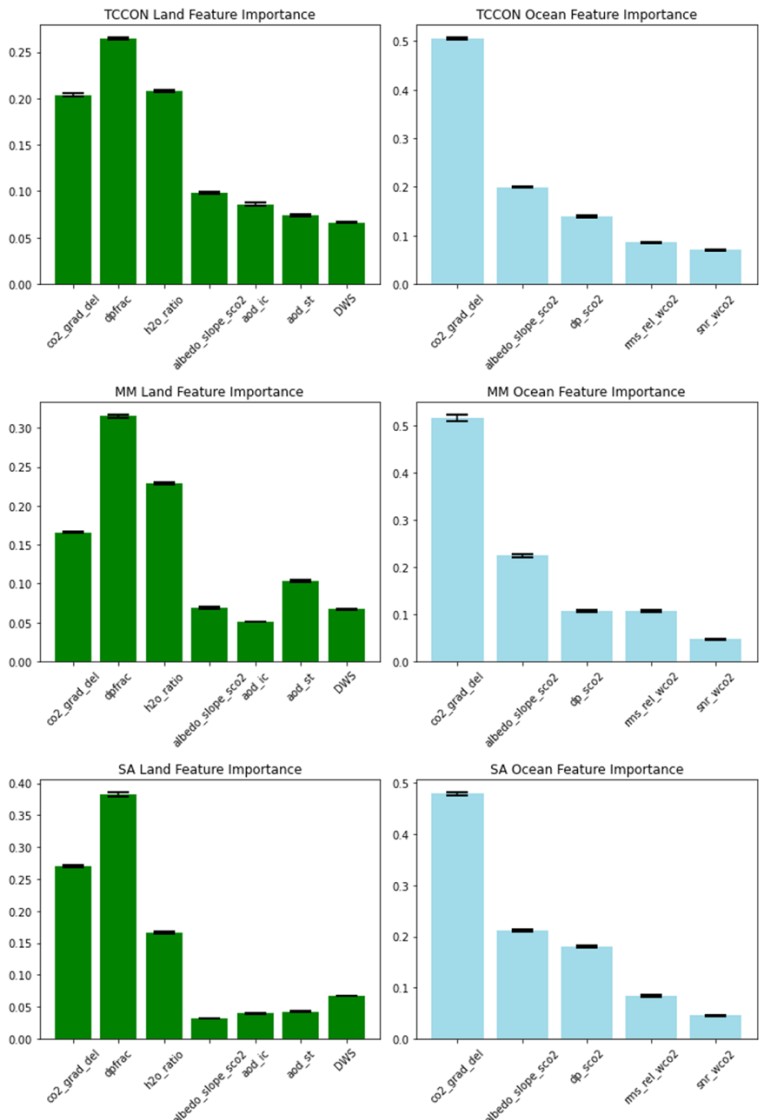

**Figure A2. Feature importance for individual truth proxy models. Error bars indicate variance over 10 runs with different random seeds.**

**Appendix B: Threshold values for QFNew**

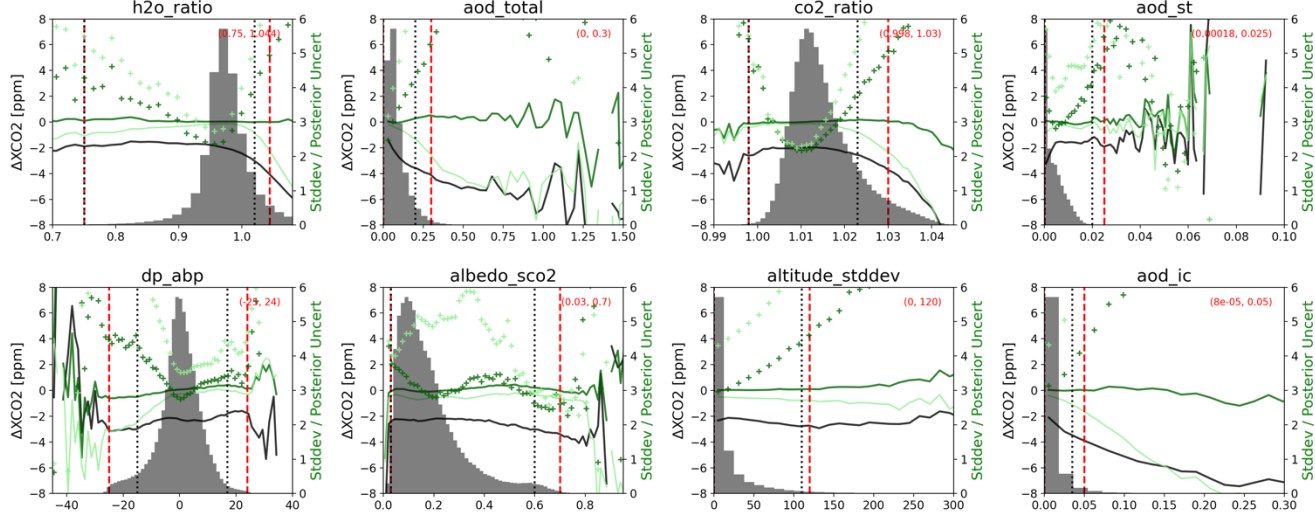


**Figure B1: Variables selected for land QFNew: the difference between the uncorrected retrieval and the model mean truth proxy is shown with the black curve. The difference between the operational correction and the model mean truth proxy is shown in the light green curve. The difference after the non-linear correction is show by the dark green curve. The binned Std error divided by the posterior uncertainty of XCO2 is show by the green pluses and right y-axis. B10 QF filters are indicated by the black vertical dashed**

**lines and QFNew is shown by the red dashed lines. Region of data denoted as QFNew=0 is contained within the red values in the parentheses.**

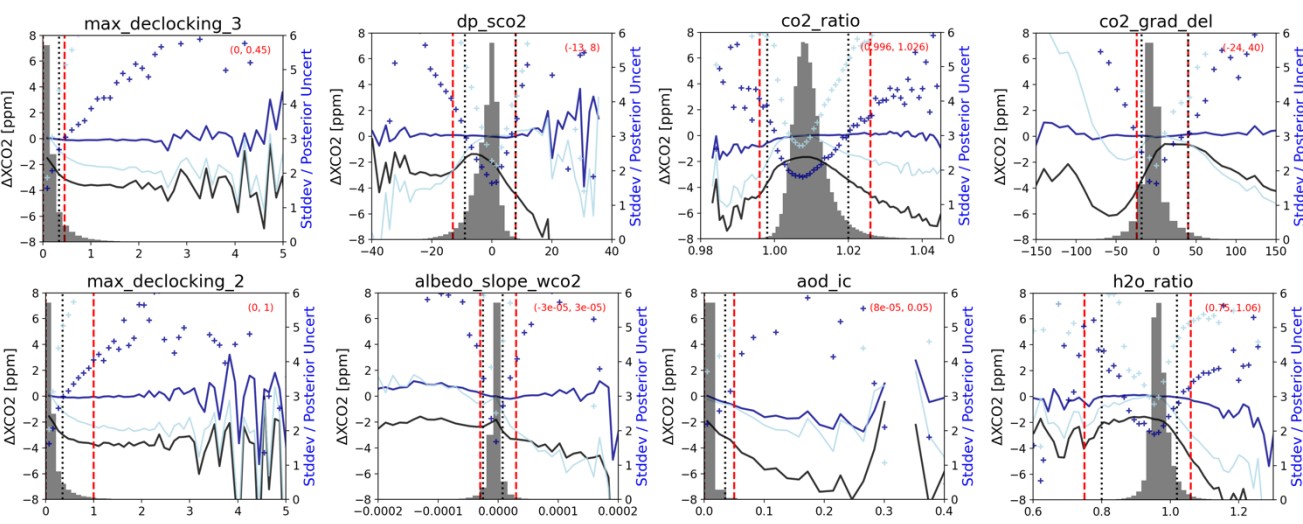

**Figure B2: Variables selected for ocean QFNew: the difference between the raw retrieval uncorrected retrieval and the model mean truth proxy is shown with the black curve. The difference between the operational correction and the model mean truth proxy is**

shown in the light blue curve. The difference after the non-linear correction is show by the dark blue curve. The binned Std error divided by the posterior uncertainty of XCO2 is show by the blue pluses and right y-axis. B10 filters are indicated by the black vertical dashed lines and a potential filter is shown by the red dashed lines. Region of data denoted as QFNew=0 is contained within the red values in the parentheses.

## Appendix C: Lite file variables

**Table C1. Features used or considered for the operational and proposed bias correction and filtering.**

| State Variable | Description | Used for: [B10 BC, ML BC, QF, QFNew] * indicates changed filter threshold for QFNew. |
|---|---|---|
| **dpfrac** | Surface pressure difference that considers smaller dry air columns over higher elevations (Kiel et al. 2019). | B10 BC, ML BC, QF, QFNew |
| **h2o_ratio** | Ratio of retrieved $H_2O$ column in weak and strong $CO_2$ bands by IMAP-DOAS. | ML BC, QF, QFNew* |
| **DWS** | Additive combination of retrieved dust, water, and sea salt aerosol optical depth. | B10 BC, ML BC, QF, QFNew |
| **aod_strataer** | Retrieved upper tropo+stratospheric aerosol optical depth at 0.755 microns. | ML BC, QF, QFNew |
| **aod_ice** | Retrieved ice cloud optical depth at 0.755 microns. | ML BC, QF, QFNew* |
| **co2_grad_del** | Difference between the retrieved vertical $CO_2$ profile and prior. | B10 BC, ML BC, QF, QFNew |
| **dp_sco2** | Surface pressure difference between the retrieved and prior, evaluated for the strong $CO_2$ band location on the ground. | B10 BC, ML BC, QF, QFNew* |
| **snr_wco2** | The estimated signal-to-noise ratio in the continuum of the weak $CO_2$ band. | ML BC, QF, QFNew |
| **co2_ratio** | Ratio of retrieved $CO_2$ column in the weak and strong $CO_2$ bands by IMAP-DOAS | QF, QFNew* |
| **altitude_stddev** | The standard deviation of the surface elevation in the target field of view. Unit is in meters. | QF, QFNew* |
| **max_declocking_wco2** | An estimate of the absolute value of the clocking error in the weak $CO_2$ band expressed as a percent. | QF, QFNew* |
| **max_declocking_sco2** | An estimate of the absolute value of the clocking error in the strong $CO_2$ band expressed as a percent. | QF, QFNew* |
| **dp_o2a** | The difference in retrieved surface pressure to $O_2A$ surface pressure prior. | QF, QFNew |
| **dp_abp** | The difference in the retrieved surface pressure to the fast $O_2A$ band pre-processor retrieval. | QF, QFNew* |

| | | |
|---|---|---|
| **albedo_slope_sco2** | Retrieved strong band reflectance slope(land) or slope of Lambertian albedo component of BRDF (ocean). | ML BC, QF, QFNew |
| **albedo_slope_wco2** | Slope of the weak $CO_2$ band albedo with respect to wavenumber. | QF, QFNew* |
| **albedo_sco2** | Surface reflectance at a reference wavelength in the strong $CO_2$ band in the primary scattering geometry from the retrieved BRDF (land). Retrieved Lambertian albedo (ocean). | QF, QFNew* |
| **albedo_quad_sco2** | Quadratic coefficient of the albedo_sco2 term with respect to wavenumber (land only). | QF, QFNew |
| **albedo_quad_wco2** | Quadratic coefficient of the albedo_wco2 term with respect to wavenumber (land only). | QF, QFNew |
| **aod_total** | Retrieved aerosol optical depth of cloud and aerosol at 0.755 microns. | QF, QFNew* |
| **rms_rel_sco2** | RMSE of the L2 fit residuals in the strong $CO_2$ band relative to the signal. | QF, QFNew |
| **rms_rel_wco2** | RMSE of the L2 fit residuals in the weak $CO_2$ band relative to the signal. | ML BC, QF, QFNew |
| **detlaT** | Retrieved offset to prior temperature profile in Kelvin. | QF, QFNew |
| **aod_sulfate** | Retrieved aerosol optical depth of sulfate aerosol at 0.755 microns. | B10 BC, QF, QFNew |
| **aod_oc** | Retrieved aerosol optical depth of organic carbon aerosol at 0.755 microns. | B10 BC, QF, QFNew |
| **aod_water** | Retrieved aerosol optical depth of water aerosol at 0.755 microns. | QF, QFNew |
| **dust_height** | Retrieved central pressure of the dust aerosol layer, relative to the retrieved surface pressure. | QF, QFNew |
| **aod_seasalt** | Retrieved aerosol optical depth of sea salt aerosol at 0.755 microns. | QF, QFNew |
| **Fs_rel** | Retrieved fluorescence relative to the $O_2A$ band continuum signal. | QF, QFNew |
| **chi2_wco2** | Reduced chi-squared value of the L2 fit residuals for the weak $CO_2$ band. | QF, QFNew |
| **windspeed** | Retrieved surface wind speed over water surfaces. | QF, QFNew |
| **water_height** | Retrieved central pressure of the cloud water layer, relative to the retrieved surface pressure. | QF, QFNew |

**Data Availability**

OCO-2 B10 Lite Files can be found at: https://doi.org/10.5067/E4E140XDMPO2 (OCO-2 Science Team et al., 2020). Proposed quality filter dataset: https://doi.org/10.17605/OSF.IO/CX53S .

**Author contribution**

WK conducted the experiments and formulated the manuscript and figures. CO prepared and provided the truth proxy data sets used. SM, SC, and CO provided significant conceptual input for experiment design and analysis of results. All authors provided thorough review and comment on the final paper.

**Competing interests**

The authors declare that they have no conflict of interest.

**Acknowledgment**

The authors would like to thank the institutions that provide data from the TCCON instruments, and to the OCO-2 algorithm
and science teams at CSU/CIRA and JPL. CarbonTracker results were provided by NOAA ESRL, Boulder, Colorado, USA, from the website at http://carbontracker.noaa.gov (last access: 10 January 2022).

**Financial support**

WK and SC were supported by the NASA GeoCarb Mission (80LARC17C0001). SM was supported by the Jet Propulsion
Laboratory, California Institute of Technology, under contract with the National Aeronautics and Space Administration (80NM0018D0004). CO was supported by a subcontract with the Jet Propulsion Laboratory.

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
