# Peer review of "A non-linear data driven approach to bias correction of XCO2 for OCO-2 NASA ACOS version 10"

_EGUsphere, 2023_

## Author Comment (AC1)

**Response to Reviewer #2**

**A non-linear data driven approach to bias correction of XCO2 for OCO-2 NASA ACOS version 10**

**William Keely et al.**

We sincerely thank the reviewer for their time in giving thorough feedback and suggestions. The points raised have led to significant changes to the manuscript. The points from the reviewer are shown in black, with our response in blue, and changes or additions to the manuscript in red.

Could the authors clarify if this understanding is correct? There are two XGBoost models (one for ocean and one for land) trained on data from all three proxy datasets that are the main contribution of this paper and these two models are used for Table 4, Table 5, Figure 3, Figure 4, Table 6, Figure 5, Table 7, Figure 7, Figure 8, and Figure 10. An additional six XGBoost models (one for each of the two surface types and three proxy datasets) are trained for Figures 2 and 9 to understand feature importance, but these models are not applied elsewhere. If this is correct (or incorrect), I believe Section 3.4 on Experiment Design could make this more clear.

This is correct, our final proposed bias correction is a land model and an ocean model trained on all three truth proxies. To clarify this, we have re-written and streamlined the Experiment Design section and throughout the manuscript.

**3.4 Experiment Design**

First, the footprint correction as described in O'Dell et al., 2018 is applied to the training and evaluation datasets. We then evaluate two methods for bias correcting retrieved $XCO_2$: a non-linear machine learning model called XGBoost; and as a baseline, we also train a MLR model similar to the hand-tuned model used in the operational correction. For correcting land nadir, and land glint data, a single XGBoost model and MLR are trained using all three truth proxies. The predictor variables, or features are the same for both model types. This allows for comparison between the non-linear model and baseline linear method to properly assess that improved fit is coming from the captured non-linearity and not just the inclusion of the additional predictors. A single XGBoost and MLR are derived for correcting ocean glint data, again using all three proxies and same set of ocean features. We also compare our approach to the operational land correction and ocean correction for B10.

To identify a set of informative features to be used as inputs for the XGBoost land and ocean models, we first train as set of models independently on each truth proxy. These six models (three for land and three for ocean) are initially fit on a large set of potentially informative features, using QF = 0 + 1 data. The resulting feature importance derived from these initial models is used to filter down the feature set to identify a subset of features that is highly informative across truth proxies. The resulting feature sets are combined to train the final proposed model pair (one for land and one for ocean), which are trained using all truth proxies.

Next, we compare the final models trained on QF = 0 + 1 data against models trained only on "good" quality data assigned QF = 0 then, evaluate each model pair on QF = 0 soundings that have been temporally held out. This is to ensure the ability of the nonlinear method to reproduce the linear model, which is the currently accepted community standard. Secondly, we evaluate the model trained on QF = 0 + 1 data on the excluded regime of data labelled QF = 1 where non-linear relationships between $\Delta XCO_2$ and predictors become more pronounced. Finally, we derive a new quality flag (QFNew) used in conjunction with the non-linear correction to that increases the throughput of well corrected data while maintaining similar error metrics as the operational filter and correction.

The authors seem to go through a lot of trouble to produce the XGBoost models for different proxy types for Figures 2 and 9, but there is little discussion other than Lines 209-213, Lines 399-401, and Lines 405-406 which take the form of "they are different for different proxy types." I suggest the authors either simplify the feature importance discussion to the XGBoost models used for the rest of the paper (trained on all proxy data and just divided by land/water) or improve the discussion related to information from different proxy datasets.

In the Feature Selection section we have simplified the wording to a more succinct explanation of the feature selection process, including a Figure 2 that now only shows the final selected variables for the proposed land and ocean models.

**4.1 Feature Selection**

[revised manuscript text omitted]

It would be very helpful to have a table (could be in the main text or an Appendix) defining all of the variables discussed in this paper. Ideally this table would contain all of the variables considered for all of your models (I believe the "subset of 27 co-retrieved state vector variables" stated in Line 202). This table would be useful for a few reasons. (1) While most variables are already defined in Table 3, some of the variables used for QFNew in Figures 7 and 8 are never defined (e.g., max_declocking_3). (2) It would be useful to have a little more information about the variables, such as how co2_grad_del is calculated (Equation 5 from O'Dell2018).

We have added Table C1 to the appendix that includes all variables considered or used in the operational and proposed bias correction and quality filters.

[revised manuscript text omitted]

We have also provided more information for co2_grad_del. We have also added clarification to section 4.1 as shown in a response farther below.

*co2_grad_del* $= [CO_{2,ret}(1) - CO_{2,ret}(0.6316)] - [CO_{2,prior}(1) - CO_{2,prior}(0.6316)]$
(4)

For the machine learning model evaluation, it is my impression only 2018 should be used (the testing dataset). It is not clear in Figures 3/4/7/8/10 what date range is being used, but it should in theory be only 2018 since the model has been trained with the data for other years and the goal is to see how generalizable the model is to data it has never seen. This is concerning for Figure 5, which tries to evaluate the model using (in part) data from 2016 and 2017 that the model was already trained on. This could suggest corrections that are overly optimistic.

We have clarified the date ranges used in figures throughout the manuscript. For Figure 5 - to increase the amount of data for plotting we use three models. Each model is used to infer bias for a validation year that is held out during model training.

**Figure 5. Remaining XCO₂ biases (ΔXCO₂) after correction for 2016-2018 and model mean proxy, binned to a 2ºx2º resolution. ΔXCO₂ after the XGBoost correction for QF=0 is shown in (a), ΔXCO₂ after the B10 correction for QF=0 is shown in (b), ΔXCO₂ after the XGBoost correction for QF=1 is shown in (c), ΔXCO₂ after the B10 correction for QF=1 is shown in (d), and difference (B10 –**

**XGB) for QF=0 is shown in (e). Three models are trained each with one year in [2016,2017,2018] used as holdout. The results on the holdout sets are then used for plotting.**

To go further with validation, have the authors considered leaving one of these proxies (like TCCON) completely out of their bias correction (or bringing in an independent dataset)? It seems to me that at the end of this, you have no independent datasets to evaluate your bias-corrected retrievals with, as they have all been used for training the model.

We now address the potential circularity induced by the lack of a truly independent proxy in section 5.1. The issue of lack of independent truth proxy in the operational correction is discussed in Taylor et al., 2023 and is still a current research focus by the OCO science team. We evaluate bias correction models trained on two of the proxies and evaluate on the third withheld proxy.

**5.1 Generalization across proxies**

We acknowledge that even with a temporal training and testing split, there is still some circularity due to the lack of a truly independent truth proxy. This issue has been discussed at length for the operational bias correction in Taylor et al. 2023 and comparison and selection independent validation data sets is still an open area of study. The risk of overfitting due to circularity become greater when fitting a more complex machine learning model. To evaluate generalizability to a fully independent validation proxy, we fit a set of XGBoost models on two truth proxies and evaluate on the third proxy which is held out during training. The same temporal split is used where 2018 data for the held-out proxy is used for evaluation. Results are shown in Figure 7, for land, and Figure 8, for ocean. Each column shows the residual fit for the hold out proxy, for QF = 0 (top row) and QF = 1 (bottom row). For QF = 0, increase in RMSE was minimal for both surface types and across proxies. There was some impact to performance on QF=1 data, when compared to training with all three proxies, particularly for TCCON with an increase in RMSE of ~0.1 ppm for land and ocean data. Indicating that the information contained in TCCON is not adequately represented by the model mean and small area approximation proxies which capture variability at larger scales. A potential approach to reducing circularity in the evaluation of the truth proxies would be to train the bias correction on TCCON and either the model mean or small area approximation, using the third proxy not chosen for validation.

[Figure]

**Figure 7. Comparison of XCO₂ derived from truth proxy (XCO₂ True) vs. XCO₂ corrected by XGBoost (XCO₂ ML) for land by hold out proxy set and hold out year (2018). Left-most column displays results of a XGBoost model trained on [TCCON, Model Mean] and evaluated on Small Area. Middle column displays results of a XGBoost model trained on [Small Area,TCCON] and evaluated on Model Mean. Right-most column displays results of a XGBoost model trained on [Model Mean, Small Area] and evaluated on TCCON. Generalization for the hold proxy and QF=0 is shown in the top row and QF=1 in the bottom.**

[Figure]

**Figure 8. Comparison of XCO₂ derived from truth proxy (XCO₂ True) vs. XCO₂ corrected by XGBoost (XCO₂ ML) for ocean by hold out proxy set and hold out year (2018). Left-most column displays results of a XGBoost model trained on [TCCON,Model Mean] and evaluated on Small Area. Middle column displays results of a XGBoost model trained on [Small Area,TCCON] and evaluated on Model Mean. Right-most column displays results of a XGBoost model trained on [Model Mean, Small Area] and evaluated on TCCON. Generalization for the hold proxy and QF=0 is shown in the top row and QF=1 in the bottom.**

Reference added: ADD TO BOTH RESPONSES

For Table 1 and the rest of the paper, what version of TCCON data is used here? Presumably GGG2014 given the reference list. Is there a reason to not use GGG2020 at this point? It is my understanding that OCO-2 B10 uses the same prior as GGG2020, so this would be more appropriate. If not, it might be necessary to account for the difference in priors when comparing GGG2014 and OCO-2 data in calculating deltaXCO2 (if this effect is large).

The operational correction for B10 is fit using GGG2014. In order to provide a faithful comparison to the operational correction we also use GGG2014. We now adequately clarify this in the manuscript.

**2.1 TCCON truth proxy**

TCCON is a system of ground-based sun-looking Fourier Transform Spectrometers with growing global coverage, that retrieve dry air mole column averaged measurements of the trace greenhouse gases from radiances in similar spectral bands to OCO-2. Since each site has been extensively validated against

WMO-traceable in situ observations aboard aircraft, TCCON offers the most accurate comparison for $XCO_2$ (Wunch et al., 2010). While TCCON is well calibrated, site coverage is limited outside of North America, Europe, and Oceania. The TCCON data set therefore is spatially the sparsest of the three truth proxies and offering non-uniform point comparisons. We use the same dataset as the operational correction consisting of OCO-2 soundings co-located TCCON GGG2014 measurements (Wunch et al., 2017; Wunch et al., 2011) in space (2.5º lat, 5º lon) and time (2h).

I am suspect of the authors' claim (Lines 65, 438) that this method is "reproducible." There is still some hand-tuning in these methods, including picking which variables to include for the regression task (How do you reconcile the different proxies saying different variables are important in Figure 2? How do you pick which redundant variables to drop based on correlation before doing the analysis in Figure 2?) and how to adjust the filters for QFNew. This is fine, but with no code published alongside the paper, this could be difficult to reproduce.

We fully agree with this statement. Subjectivity is still largely present in the selection of variables and hand tuning of filter thresholds. We therefore only claim that the framework can be adapted to future algorithm updates.

Is there a plan to incorporate this into future versions of the OCO-2 data? Regardless, will the authors be making available the bias-corrected data produced in this paper?

A machine learning bias correction is planned for a future lite file update for B11 that expands on the approach discussed in this work. While there are currently no plans for a machine learning correction for B10 we plan on providing the data used in this paper.

**Specific Comments**

Line 59: I am not sure if "relative to the operational linear correction" is accurate with respect to the TROPOMI methane retrieval in Schneising et al. (2019).

Line 61: a slightly longer discussion of how this work differentiates from Mauceri2023 could be appropriate.

Line 59-61: We now address both comments in the paragraph.

A drawback of applying the quality filter is the exclusion of data due to the linear assumption of the bias correction to which the quality filter limits the regime of interaction between state vector variables and $\Delta XCO_2$. Due to loss of data, the bias correction and quality filter are often disregarded for local studies (Nassar et al., 2017; Mendonca et al., 2021) or too limiting for certain regions (Jacobs et. al., 2020). Applying non-linear machine learning techniques have shown great promise for the task of bias correction for GOSAT/GOSAT-2 (Noël et al., 2022) and TROPOMI (Schneising et al., 2019). Specific correction of 3D cloud biases for OCO-2 retrieved $XCO_2$ (Massie et al. 2016) using a non-linear method fit on a small set of features correlated with 3D cloud effects in addition to the linear operational correction, is demonstrated in Mauceri et al. (2022).

Line 70: are the averaging kernels taken into account when comparing OCO-2 to TCCON or the model atmospheres?

The averaging kernels are used for both the TCCON truth proxy and concentration fields from the flux models proxy. We have added an additional citation to the paragraph that further explains the application of the OCO-2 averaging kernel in the derivation of the truth proxies.

To develop a bias correction, we define three truth proxy data sets for the true atmospheric column mode fraction. $\Delta XCO_2$ is then set as the difference between the raw ACOS retrieval of $XCO_2$ and the truth proxy estimate of $XCO_2$ as shown in Eq. 1. For the TCCON and model mean truth proxies, the OCO-2 kernel is also applied as described in Taylor et al., 2023.

Figure 1: The "N = 1022636" title for Figure 1a seems to be a typo. I would expect the number of soundings here to be close to the total number of OCO-2 soundings since model grids are continuous in space and time (depending on how often the 1.5 ppm threshold is passed) and thus N for the model proxy should be > N for TCCON, not equal. On a related note, what is the total number of OCO-2 soundings considered to give a sense of the percentage used for each proxy dataset?

We have corrected the OCO-2 sounding count in the figure. The soundings come a quick test set (QTS) which is approximately 5% of the full OCO-2 record as described in Taylor et al., 2023.

[Figure]

**Figure 1: Spatial coverage for each truth proxy. The mean of a set of flux models is shown in (a), small area approximation is shown in (b), and TCCON is shown in (c).**

The soundings considered come a quick test set (QTS) which is approximately 5% of the full OCO-2 record as described in Taylor et al., 2023.

Table 1 contains errors for the Tsukuba altitude, Sodankylä altitude, Izaña altitude, Wollongong latitude, Réunion latitude, Lamont latitude, and Karlsruhe latitude (and maybe others).

Table 1 has now been corrected.

Table 2: could you specify the resolutions of these models?

Table 2 now includes an additional column that with spatial and temporal resolution.

**Table 2. Flux models used for the model mean truth proxy. TM5 – Transport model 5, TM3 – Transport model 3, LMDZ – Laboratoire de Meteorology, EnKF – Ensemble Kalman Filter, 4D-Var – 4 Dimensional Variation.**

| Model name | Institute | Transport model | Resolution [latxlonxtime] | Inverse method | Citation |
|---|---|---|---|---|---|
| CarbonTracker | NOAA Global Monitoring Laboratory | TM5 | 2º×3º×3h | EnKF | Peters et al. (2007) CarbonTracker (2021) |
| CarboScope | Max Planck Institute for Biogeocehmistry | TM3 | 4º×5º×6h | 4D-Var | Rödenbeck (2005); Rödenbeck et al. (2018) CarboScope (2021) |
| CAMS | Copernicus Amosphere Monitoring Service | LMDZ | 1.9º×3.75º×3h | 4D-Var | Chevallier et al. (2010) CAMS (2021) |

Line 128-129: it is my understanding (and you state as such in line 130) that XGBoost is not the average across an ensemble, but rather the sum across the ensemble.

Yes, thank you for catching this error. We have corrected this in the manuscript.

Line 135: how do you search for these hyperparameters? Are these the same for all of the XGBoost models discussed in this paper?

We now thoroughly explain the hyperparameter search and regularization values for the final proposed bias correction models in the paragraph.

XGBoost employs $L_1$ and $L_2$ norm regularization to reduce overfitting to outliers present in the training dataset. The effect of the regularization is governed by the hyper-parameters $\lambda$ and $\gamma$, and must be carefully selected or tuned. To find these hyper-parameters we use a k-fold cross validation strategy in which the training dataset is divided into $k$ subsets (we use $k=10$) and each subset is sequentially held out for evaluation for a model trained on the rest of the data. Performance across the k-folds is averaged and the process is repeated for each potential selection of hyper-parameters. We found a $\lambda_{LAND}=2.5$ and $\gamma_{LAND}=3.75$ for the land correction, and $\lambda_{OCEAN}=2.0$ and $\gamma_{OCEAN}=10.0$.

Line 179: Is data from 2014-2018 used for all three proxy datasets?

Yes, this is correct. Data range for all three proxies is from 2014-2018.

Line 184-185: In Section 3.3 and elsewhere, it is stated that the models are trained for Ocean G and Land NG (2 models). In these lines, it is suggested that there are three models.

We agree the final proposed correction is not clear and the section has been reworded. We have also worked to clarify this throughout the manuscript.

**3.3 Training and test split**

For training and evaluating the non-linear correction, we subset each of our truth proxy datasets into a training and testing datasets. First, datasets are split by the two surface types: ocean and land. In the B10, both operation modes (nadir and glint) are combined for the land bias correction due to low variance in feature importance between nadir and glint (O'Dell et al. 2018). To compare to the operational correction, we also combine both modes for the land correction model. The land and ocean data sets are subset once more by truth proxy to identify informative features for the final land and ocean models. To ensure that model performance is indicative of how well the models generalize to unseen data, we hold out a year of data for evaluation of the final land model and ocean model. Models are trained on data from 2014, 2015, 2016, and 2017, then evaluated on data from 2018.

Line 206: Is there a threshold you used for the correlation coefficient?

We now clarify this value in the revised paragraph.

To ensure robustness to correlation among features we (which information gain does not account for) we calculate Pearson's correlation values between features. Features with an absolute Pearson value greater than 0.5 are included one a time and the feature with the highest importance is kept.

Line 219: Is there a reason to specify that the prior pressure is from the strong band? Is the prior pressure different for the weak band?

Yes, this due to an alignment offset in the pointing location between the three bands and there for the priors are used.

Line 233: Why is the variable dp_sco2 considered? Lines 220-221 dp_frac discussed the disadvantages of this kind of pressure difference term.

dpfrac takes into account the surface elevation and is only available over land, while dp_sco2 is used for ocean glint scenes. We have clarified this in the Section 4.1:

**4.1 Feature Selection**

[revised manuscript text omitted]

Figure 2: Why are there a different number of considered features for land and ocean? Were the same 27 variables (Line 202) started with and a different subset dropped for LandNG versus OceanG because of different correlations (Line 206) for the different operation modes?

Some features are not available or held constant over the ocean. We have simplified Figure 2 to show only the final feature importance of the proposed correction models, as seen above.

Line 276: what percentage of retrievals are filtered because of this?

h2o_ratio filters out ~10% of the soundings.

Figures 3 and 4: Is this just 2018 data? Could you add arrows to indicate the direction of the filters (or write the ranges like in Figures 7 and 8)?

We have added arrows to indicate the region assigned QF=0 or "good" quality data.

[Figure]

**Figure 3. ΔXCO₂ vs land features for 2018. Mean interaction and 2σ Stddev for uncorrected ΔXCO₂ plotted in red, XGBoost corrected in green and B10 corrected in blue. The vertical black dotted lines indicate B10 QF filters and arrows point towards the region assigned QF=0. Individual soundings are shown with grey scatter.**

[Figure]

**Figure 4. ΔXCO₂ vs ocean features for 2018. Mean interaction and 2σ Stddev for uncorrected ΔXCO₂ plotted in red, XGBoost corrected in green and B10 corrected in blue. The vertical black dotted lines indicate B10 QF filters and arrows point towards the region assigned QF=0. Individual soundings are shown with grey scatter.**

Figure 5: It seems to me you cannot properly evaluate the model with the training data (2016-2017) and this plot should only show 2018 data. Additionally, I am interested to know if this is data from all proxy datasets? This would help answer the question of if the remaining differences are due to shortcomings in the bias correction method or in the proxy datasets.

Thank you for raising these points, this plot needed much clarification. The biases for each year in 2016-2017 are estimated by a model (3 in total) trained on all years except for one year within the range which was used for inference. We used this process to increase the amount of available data for plotting – only the model mean is plotted but is included in for training. To address generalization to a held proxy we now have added a new Section 5.1 as shown above.

[Figure]

**Figure 5. Remaining XCO₂ biases (ΔXCO₂) after correction for 2016-2018 and model mean proxy, binned to a 2ºx2º resolution. ΔXCO₂ after the XGBoost correction for QF=0 is shown in (a), ΔXCO₂ after the B10 correction for QF=0 is shown in (b), ΔXCO₂ after the XGBoost correction for QF=1 is shown in (c), ΔXCO₂ after the B10 correction for QF=1 is shown in (d), and difference (B10 – XGB) for QF=0 is shown in (e). Three models are trained each with one year in [2016,2017,2018] used as holdout. The results on the holdout sets are then used for plotting.**

Line 338: Is this trained on QF = 0 + 1 data and all three of the truth proxies? And two different models (one for land and one for ocean)?

Yes, this is correct. We have clarified that in the manuscript.

**4.5 Increased sounding throughput**

One of the benefits of the non-linear bias correction is the potential for increased throughput of well-corrected QF = 1 data. Improved throughput of well corrected data would be of benefit to point analysis

studies where data is limited by the operational QF, and potentially of benefit to flux models as well. To provide an empirical example of this, we create a modified version of the operational $XCO_2$ quality flag utilizing our proposed ocean correction model and land correction model. We take a conservative approach where initial filter values are set equal to those of the operational quality filtering. Then, we select a few variables for which the filters are relaxed to increase sounding throughput while maintaining the RMSE of the combined operational correction and quality filter. With our new quality flag (QFNew), we are able to increase sounding throughput by approximately **16%** over the B10 QF while matching the RMSE of the B10 correction as shown in Table 5.

Figures 7/8: It is not clear to me what each of the colors represent. The black line is deltaXCO2 for the raw XCO2 retrievals. But between Line 348 and the figure captions, I can't figure out what the difference between the light and dark green/blue lines are (maybe dark is XGBoost bias-corrected and light is operationally bias-corrected?).

We have added clarification to the figure text. Figure 7 & 8 have now been moved to the appendix in order to stream line the section.

[Figure]

**Figure B1: Variables selected for land QFNew: the difference between the uncorrected retrieval and the model mean truth proxy is shown with the black curve. The difference between the operational correction and the model mean truth proxy is shown in the light green curve. The difference after the non-linear correction is show by the dark green curve. The binned Std error divided by the posterior uncertainty of XCO2 is show by the green pluses and right y-axis. B10 QF filters are indicated by the black vertical dashed lines and QFNew is shown by the red dashed lines. Region of data denoted as QF=0 is contained within the red values in the paratheses.**

[Figure]

**Figure B2: Variables selected for ocean QFNew: the difference between the raw retrieval uncorrected retrieval and the model mean truth proxy is shown with the black curve. The difference between the operational correction and the model mean truth proxy is shown in the light blue curve. The difference after the non-linear correction is show by the dark blue curve. The binned Std error divided by the posterior uncertainty of XCO2 is show by the blue diamonds and right y-axis. B10 filters are indicated by the black vertical dashed lines and a potential filter is shown by the red dashed lines. Region of data denoted as QF=0 is contained within the red values in the paratheses.**

Figures 5/10: titles or different labels on the colormaps might make it more clear which plots are for operationally bias-corrected data and which are for XGBoost bias-corrected data (but this is clear in the caption).

We agree and have added titles as seen above.

Figure 10: Is this just for 2018? Is it for all proxy datasets?

The figure shows 2018 data, this is now clarified in the figure text. Note, Figure 10 is now Figure 11.

**Figure 11. Hex bin plots show conditional distributions of 2018 ΔXCO₂ vs. dpfrac and h2o_ratio. Remaining ΔXCO₂ after the operational correction for B10 is shown in (a). Remaining ΔXCO₂ after the non-linear correction is shown in (b). Binned sttdev of ΔXCO₂ divided by the posteriori uncertainty from the retrieved X$_{CO2}$ is shown in (c) for the operational correction for B10 and (d) for the non-linear correction. B10 QF filter thresholds for both features are shown with black dashed lines for reference.**

There are in-text references for Kuze2009, Palmer2019, Crowell2019, Peiro2021, Mendonca2021, Jacobs2020, Osterman2020, Worden2017, Taylor2012, Morino2018a, Morino2018b, and Hase2015 (and maybe others) in the text and tables that are missing in the Reference section.

Thank you for thoroughly checking for and catching these errors. We have now corrected these in the revised manuscript.

**Technical Corrections**

There are many typos throughout this paper. It needs to be significantly cleaned up before publication. I tried to catch as many as I could here.

Line 12: Obersvatory => Observatory

Corrected.

Line 15: correlate => correlated

Corrected.

Line 35: m). => m.

Corrected.

Line 36: Rogers => Rodgers

Corrected.

Line 42: Missing subscript on XCO2

Corrected.

Line 47: emperically => empirically

Corrected.

Line 53: reduce => reduces

Corrected.

Line 55: filter => filters

Corrected.

Line 55: correction. to which => correction to which

Corrected.

Line 57: missing word before "or too limiting"

Or is too limiting

Line 59: 2021 => 2022

Corrected.

Line 65: reproduceable => reproducible (or be consistent with Line 438)

Claims of reproducibility have been removed.

Line 66: remove "upcoming missions such as GeoCarb"

This has now been removed.

Line 68: mode => mole

Corrected.

Line 78: Lever => Level

Corrected.

Line 86: remove comma

Removed

Line 91: offering => offers

Corrected.

Table 1 caption: rephrase/missing words in "proxy the TCCON"

**Table 1. TCCON sites used in bias correction and filtering for B10 ACOS.**

Line 104: missing subscript on XCO2

Corrected.

Line 106: XCO2, is => XCO2 is

Removed.

Line 106: overs => offers

Corrected.

Line 110: Table 1 => Table 2

Corrected.

Line 121: employee => employ

Corrected.

Line 132: into => in

Removed.

Line 134: we hold out small subset => we hold out a small subset

Removed.

Line 145: variables not defined (though common notation is used)

The variables in Eq 3 are now explained in the text.

Line 172: remove "a"

Removed.

Line 176: mode; => mode,

Corrected.

Line 185: or features are => or features, are

Removed.

Line 186: This, allows =? This allows

Removed.

Line 192: data then, => data and then

Removed.

Line 205: delete "we"

Removed.

Line 213: Table 1 => Table 3

Corrected.

Line 223: retrivals => retrievals

Corrected.

Line 227 (and Table 3): aod_stratear => aod_strataer

Corrected.

Line 232: In addition to the albedo_slope_sco2, four => In addition to albedo_slope_sco2 and co2_grad_del, three

Removed.

Table 3 caption: features for in => features for use in

Corrected.

Table 3: dpfrac versus dp_frac is inconsistent between the text (Line 216) and here/other figures.

Corrected throughout to dpfrac to stay consistent with lite file variable naming.

Section 4.1 is repeated (4.1 Feature Selection, 4.1 Model evaluation for QF = 0).

Corrected.

Line 261: Table 2 => Table 4

Corrected.

Line 275: Sentence beginning with "However," could be rephrased. The portion inside parentheses is confusing.

Sentence now reads:

Variables such as h2o_ratio which are responsible for the bulk of the quality filtering (h2o_ratio thresholds remove ~10% of soundings) exhibit such non-linear characteristics over their marginal distributions.

Line 280: Table 3 => Table 5

Corrected.

Figures 3/4: inconsistent x-axis labels with the text/Table 3 (e.g., aod_st versus aod_strataer). Arrows designating the direction of the filters (e.g., on the aod_st plot) could be helpful but are not necessary.

[Figure]

**Figure 3. ΔXCO₂ vs land features for 2018. Mean interaction and 2σ Stddev for uncorrected ΔXCO₂ plotted in red, XGBoost corrected in green and B10 corrected in blue. The vertical black**

[Figure]

**Figure 4. ΔXCO₂ vs ocean features for 2018. Mean interaction and 2σ Stddev for uncorrected ΔXCO₂ plotted in red, XGBoost corrected in green and B10 corrected in blue. The vertical black dotted lines indicate B10 QF filters and arrows point towards the region assigned QF=0. Individual soundings are shown with grey scatter.**

Line 304: Table 4 => Table 6

Corrected.

Line 307: double-check percentages. For example, for ocean QF=0, ((0.67^2)-(0.61^2))/(0.67^2) = 17%, not 4% (if I correctly understand your methodology).

Thank you for catching this. We have updated these values.

Table 6: stddev => standard deviation; caption mentions raw XCO2 data, but this is not present in the table.

Raw retrieval values are now correctly included in Table 6.

Line 342: Table 5 => Table 7

Corrected.

Line 348: Figure 8 and 9 => Figures 7 and 8

Corrected to Figure B1 and Figure B2.

Line 354: benefit of quality => benefit of a quality

Corrected.

Table 7: Region/Truth Proxy => "Surface/Mode" (?); through put => throughput

Corrected.

Line 383: Qf => QF

Corrected.

Line 384: Figure 9 => Figure 10; Feaures => Features

Corrected.

Line 394: filter bound h2o_ratio => filter bounds, h2o_ratio

Corrected.

Line 400: noteably => notably

Corrected

Lines 415-416: Figure 11 => Figure 10

Figure 10 has now become Figure 11

Line 442: remove ; and rephrase

Corrected.

Line 449: ISS not defined in text

ISS is now defined correctly before abbreviation.

Title: data driven => data-driven. Hyphen usage should be reviewed throughout (Lines 12, 13, 19, 32, 52, 58, 87, 89, 103, 157, 164, 273, 336, 355, 421, 438, 447, etc.).

Corrected throughout the manuscript.

---

## Author Comment (AC2)

**Response to Reviewer #1**

**A non-linear data driven approach to bias correction of XCO2 for OCO-2 NASA ACOS version 10**

**William Keely et al.**

We sincerely thank the reviewer for their time in giving thorough feedback and suggestions. The points raised have led to significant changes to the manuscript. The points from the reviewer are shown in black, with our response in blue, and changes or additions to the manuscript in red.

In my opinion, the underlying machine learning method XGBoost is praised beyond measure (highly interpretable compared to other machine learning algorithms, improved predictive performance compared to Random Forest, highly robust to overfitting, ...). It is not trivial to prove such universal statements, and in the context of this paper it is not even necessary. On the other hand, some of these aspects are not sufficiently dealt with regarding the specific example of bias correction presented here. In this sense, please avoid questionable general statements and elaborate on the available results (the proposed bias correction) instead: How can the actually obtained model be interpreted? What is the most appropriate way to calculate feature importances aiming at maximising interpretability of the specific model in question? What is the ranking of the most important features (for land and ocean data)? Does the model overfit for the chosen parameters?

We fully agree with this statement and have removed the broader claims related to the method. We have added our full responses and manuscript changes to the in-depth points and comments below.

In order to assess (and exclude) potential overfitting tendencies, it might also be useful to define more challenging training and validation data sets or to use completely independent data for validation. The models are trained on data from 2014-2017 and are then evaluated on data from 2018. However, the validation data set is not entirely independent as the biases are likely similar in the statistical sense from year to year. It would be more instructive to leave out whole regions and/or proxy data sets in the training and only use them for validation. It would be interesting whether the figures and tables demonstrating the performance of the correction would change significantly as a result (e.g. Figures 3/4 or Table 6). For example, Figure 5 suggests that the correction does not generalise so well to regions that were rarely considered during training (e.g tropics, parts of South Asia and Canada, compare to Figure 1).

We agree with the need to assess how a machine learning bias correction will generalize to a truly independent data set. An additional section and new Figure 7 have been added to the discussion exploring the ability of the bias correction to generalize to a held-out truth proxy. We would like note that the temporal training/validation split employed in the manuscript is an improvement over the split used by the operational correction which fits on all proxies for a subset of the record and evaluates on data from the same years. Addressing the circularity in the current use of the truth

proxies and identifying new proxies for validation is an ongoing area of study by the OCO science team.

**5.1 Generalization across proxies**

We acknowledge that even with a temporal training and testing split, there is still some circularity due to the lack of a truly independent truth proxy. This issue has been discussed at length for the operational bias correction in Taylor et al. 2023 and comparison and selection independent validation data sets is still an open area of study. The risk of overfitting due to circularity become greater when fitting a more complex machine learning model. To evaluate generalizability to a fully independent validation proxy, we fit a set of XGBoost models on two truth proxies and evaluate on the third proxy which is held out during training. The same temporal split is used where 2018 data for the held-out proxy is used for evaluation. Results are shown in Figure 7, for land, and Figure 8, for ocean. Each column shows the residual fit for the hold out proxy, for QF = 0 (top row) and QF = 1 (bottom row). For QF = 0, increase in RMSE was minimal for both surface types and across proxies. There was some impact to performance on QF=1 data, when compared to training with all three proxies, particularly for TCCON with an increase in RMSE of ~0.1 ppm for land and ocean data. Indicating that the information contained in TCCON is not adequately represented by the model mean and small area approximation proxies which capture variability at larger scales. A potential approach to reducing circularity in the evaluation of the truth proxies would be to train the bias correction on TCCON and either the model mean or small area approximation, using the third proxy not chosen for validation.

[Figure]

**Figure 7. Comparison of XCO₂ derived from truth proxy (XCO₂ True) vs. XCO₂ corrected by XGBoost (XCO₂ ML) for land by hold out proxy set and hold out year (2018). Left-most column displays results of a XGBoost model trained on [TCCON, Model Mean] and evaluated on Small Area. Middle column displays results of a XGBoost model trained on [Small Area,TCCON] and evaluated on Model Mean. Right-most column displays results of a XGBoost model trained on [Model Mean, Small Area] and evaluated on TCCON. Generalization for the hold proxy and QF=0 is shown in the top row and QF=1 in the bottom.**

[Figure]

**Figure 8. Comparison of XCO₂ derived from truth proxy (XCO₂ True) vs. XCO₂ corrected by XGBoost (XCO₂ ML) for ocean by hold out proxy set and hold out year (2018). Left-most column displays results of a XGBoost model trained on [TCCON,Model Mean] and evaluated on Small Area. Middle column displays results of a XGBoost model trained on [Small Area,TCCON] and evaluated on Model Mean. Right-most column displays results of a XGBoost model trained on [Model Mean, Small Area] and evaluated on TCCON. Generalization for the hold proxy and QF=0 is shown in the top row and QF=1 in the bottom.**

There are quite a few different XGBoost models, e.g. for testing purposes, in the paper; I am not entirely sure what the proposed bias corrected product is in the end. I suspect it is the data set with the correction (split according to land and ocean) learned on all three proxy datasets for QF=0+1 simultaneously and subsequently restricted to QFNew. Is this correct? Or is it some kind of average of the three models for the different truth proxy data sets? Please make this more clear in the text.

This is correct, we propose a XGBoost model for land data and a XGBoost model for ocean data. We then derive QFNew using the new bias correction. This final proposed bias correction was not clear in the text. We have clarified the language in Section 3.4 and it now reads:

Two methods are compared for bias correcting retrieved $XCO_2$: a non-linear machine learning model called XGBoost; and as a baseline, we also train a MLR model similar to the hand-tuned model used in the operational correction. For correcting land nadir, and land glint data, a single XGBoost model and MLR are trained using all three truth proxies. The predictor variables, or features are the same for both model types. This allows for comparison between the non-linear model and baseline linear method to properly assess that improved fit is coming from the captured non-linearity and not just the inclusion of the additional predictors. A single XGBoost and MLR are derived for correcting ocean glint data, again using all three proxies and same set of ocean features. We also compare our approach to the operational land correction and ocean correction for B10.

To identify a set of informative features to be used as inputs for the XGBoost land and ocean models, we first train as set of models independently on each truth proxy. These six models (three for land and three for ocean) are initially fit on a large set of potentially informative features, using QF = 0 + 1 data. The resulting feature importance derived from these initial models is used to filter down the feature set to identify a subset of features that is highly informative across truth proxies. The resulting feature sets are combined to train the final proposed model pair (one for land and one for ocean), which are trained using all truth proxies.

Next, we compare the final models trained on QF = 0 + 1 data against models trained only on "good" quality data assigned QF = 0 then, evaluate each model pair on QF = 0 soundings that have been temporally held out. This is to ensure the ability of the nonlinear method to reproduce the linear model, which is the currently accepted community standard. Secondly, we evaluate the model trained on QF = 0 + 1 data on the excluded regime of data labelled QF = 1 where non-linear relationships between $\Delta XCO_2$ and predictors become more pronounced. Finally, we derive a new quality flag (QFNew) used in conjunction with the non-linear correction to that increases the throughput of well corrected data while maintaining similar error metrics as the operational filter and correction.

Additional wording has been added to the abstract.

In this paper, we demonstrate a clear improvement in the reduction of error variance over the operational correction by using a set of non-linear machine learning models, one for land and one for ocean soundings.

We demonstrate an approach for selecting co-retrieved state vector variables and other features to be used as input into a land model and an ocean model to correct biases in ACOS retrieved $XCO_2$.

The bias correction (for both land and ocean) includes features measuring the deviation of the retrieved quantities from the prior (surface pressure and vertical $CO_2$ profile). I assume that these priors are very consistent with the $XCO_2$ truth used in the supervised learning. Doesn't that run the risk of essentially pushing the $XCO_2$ results back to the truth/prior in specific cases without noticing in the averaging kernels that hardly any information is gained from the actual measurement? Can you exclude that point sources that are not present or not sufficiently resolved in the truth are artificially attenuated or corrected away in unseen data? Due to these potential pitfalls, the results would be more robust if such parameters (co2_grad_del, dpfrac, dp_sco2) were not used in the bias correction. How important are these features in your machine learning model? Please discuss these issues in the manuscript.

The $CO_2$ gradient delta, and surface pressure delta terms are used extensively in the operational bias correction and an analysis of the correction using these terms is given in Kulawik et al. 2019. Despite this we agree that there is potentially a risk in over correcting using the more complex machine learning model. We believe that we show that we are robust to this due to the negligible improvement in performance for QF=0 data over the operational correction, where co2_grad_del, dp_sco2, and dpfrac dominate the feature importance. When correcting QF=1 data the feature importance for these terms drops off dramatically as shown in the new Figure 9. Furthermore, we offer an empirical example using a set of plumes not present in the training data (a land example and an ocean example) to illustrate that the method does not remove $CO_2$ enhancements.

**5.3 Preservation of $CO_2$ enhancements**

We assess the risk of the proposed bias correction to correct out and remove plume features in the data. Several features heavily utilized by the XGBoost models and in operational correction such as the $CO_2$ gradient delta, and surface pressure terms (e.g., dpfrac, dp_o2a), are differences between the ACOS retrieved state, and the prior. Therefore, there is potentially a risk for the bias correction to use the delta terms to over correct the retrieved $XCO_2$ to the truth. We compare XGBoost corrected $XCO_2$ for two known plumes first identified in Nassar et al. 2021. The two example plumes are shown in Figure 10, an ocean glint plume in Taean, South Korea, and a land nadir plume observed over two co-located power plants in Ohio, US. We compare the uncorrected $XCO_2$ retrieval (B10 Raw), the operationally corrected $XCO_2$ (B10 Corrected) and the machine learning corrected $XCO_2$ (XGBoost Corrected) and note that the machine learning corrected product captures enhancements not present in the training data. These results are also consistent with the findings in Mauceri et al. 2023, that also showed fitting a machine learning model for 3D cloud correction, which include similar delta terms, did not correct out $CO_2$ enhancements.

[Figure]

(a) Taean

(b) Gavin-Kyger

**Figure 10. Two CO₂ plumes captured downwind from power plants (Nassar 2021). An ocean glint plume at Taean, South Korea, [lat 36.91º, lon 126.23º] on 2015-04-17 is shown in (a). A land nadir plume near the J. M. Gavin and Kyger Creek power plants in Ohio, USA, [lat 38.93º, lon -82.12º] on 2015-07-30. Regions with the example plumes are not present in the training dataset and consist of QF = 0 + 1 data.**

L25-29: There were other satellites measuring $CO_2$ before GOSAT, e.g. AIRS or SCIAMACHY. GOSAT is considered the first satellite designed specifically for the purpose of measuring atmospheric $CO_2$ from space. Please be more specific here.

We now correctly acknowledge the contribution of these instruments. The sentence now reads:

Following a long history of critical in situ measurements of $CO_2$ at key sites around the world that allowed us to better understand the carbon cycle on continental scales, the era of space-based remote sensing began with the Scanning Imaging Absorption spectrometer for Atmospheric Chartography (SCIAMACHY) in March 2002 (Boevnsmann et al, 1999) and the Atmospheric Infrared Sounder (AIRS) launched in May 2002 (Aumann et al, 2003).

L58-60: Do you mean (Noel et al., 2021) or (Noel et al., 2022)? There are two papers, but only (Noel et al., 2022) is in the References. (Noel et al., 202?) and (Schneising et al., 2019) both use non-linear bias correction techniques but there are no explicit comparisons to linear corrections and the term "operational" does not fit here either. Please revise this sentence.

The citation has been corrected and is no longer used to argue against a linear correction.

Applying non-linear machine learning techniques have shown great promise for the task of bias correction for GOSAT/GOSAT-2 (Noël et al., 2022) and TROPOMI (Schneising et al., 2019).

L63: Please be more specific what you mean by "interpretable" since XGBoost is a complex black-box model, which is not intrinsically interpretable. Do you mean post-hoc model interpretation methods? Do you refer to global explainability of the model or to local explainability of individual predictions?

We agree that the quality of the post-hoc interpretability provided by the internal splitting criterion was overstated. We have refined the section and clarified that the information gain provides only a global explanation of feature importance. We have clarified this in the text.

This research demonstrates a general non-linear bias correction approach for OCO-2 build 10 (B10, Taylor et al., 2023) via a machine learning method and provide a post-hoc explanation of the overall contribution of the selected state vector features.

L65: "reproducible" is somewhat misleading because it gives the impression that a universal recipe, that can be transferred without any adaptations, is being presented. However, when using other data sets the parameters of the model have to be re-tuned and the used features have to be adapted. Is there a systematic approach, e.g. to feature selection or setting of model parameters, justifying the designation "reproducible"?

We agree that the claim of reproducibility is misleading and have clarified that the general frame work for developing the bias correction can be adapted for future ACOS updates.

The framework presented in this manuscript for identifying informative features for bias correction can be adapted for future OCO-2,3 ACOS algorithm updates.

L66: The GeoCarb mission was cancelled, please remove.

Removed.

L85-92: Please specify which TCCON version you are using (GGG2014 or GGG2020?). It seems to be GGG2014, why didn't you use the most recent version? Does it make any difference in performance when training or validating with one or the other version?

Both the operational correction and our machine learning correction use the same set of soundings co-located with TCCON GGG2014 measurements. A bias correction is currently being developed for GGG2020, however that work is still unpublished. We have clarified our use of GGG2014 data.

We use the same dataset as the operational correction consisting of OCO-2 soundings co-located TCCON GGG2014 measurements (Wunch et al., 2017; Wunch et al., 2011) in space (2.5º lat, 5º lon) and time (2h).

L122-L124: I disagree with the statement that XGBoost is highly interpretable compared to other machine learning algorithms; I would even argue that XGBoost is typically the least interpretable well-established machine learning algorithm of all after deep neural networks and also requires post-hoc approaches to understand and explain the model. For example, Support Vector Machines or Random Forest are typically more interpretable than XGBoost. Please revise (or remove) the sentence accordingly.

We have removed the general statements regarding model interpretability comparisons.

L128-129: XGBoost does not average across the ensemble (in contrast to Random Forest). Instead, XGBoost trains each subsequent model in the ensemble to improve upon the errors of the previous models and uses a weighted sum of the individual model predictions to make its final prediction. This is also one of the reasons why XGBoost is usually less interpretable than Random Forest.

This has now been corrected in the sentence.

L130: It depends on the specific task, the available resources, and data set whether XGBoost or Random Forest performs better, as they have different strengths and weaknesses. Moreover, it is hard to do a fair comparison anyway, because you have to tune the corresponding parameters independently, e.g. it makes little sense to fix certain tree structures in a comparison. Therefore, please avoid a general statement comparing the predictive performance of different machine learning algorithms, especially when it comes to XGBoost and Random Forest, which often perform quite similarly after respective optimal parameter tuning (see also general comments). Have you tried other machine learning algorithms for the specific task presented here?

L131-133: The arbitrary application of these strategies does not make XGBoost per se highly robust against overfitting to the training data. Please make clearer that these strategies can avoid overfitting if the parameters and the validation data set are chosen appropriately. Please demonstrate explicitly that there is no overfitting for the non-linear bias correction presented here.

L134-135: How exactly were these parameters determined? By monitoring the performance of the model on a validation dataset during the training process and stopping the training when the performance on the validation dataset does not improve for a given number of consecutive iterations? Are the parameters actually always the same (for all testing models and the final bias correction)?

We agree and have re-written Section 3.1, removing the general comparison on model performance and clarified that regularization alone does not guarantee good generalization to an unseen data set. We have described thoroughly the hyper-parameter tuning process and have further elaborated on the parameters selected for the land and ocean models. The section now reads:

**3.1 Gradient boosting**

To model systematic error from co-retrieved state vector elements, we employee a machine learning method known as extreme gradient boosting or XGBoost (Chen et al. 2016) which can fit both linear and non-linear relationships. XGBoost is an ensemble model where a set of simple models known as regression trees (Breiman 1984) are sequentially trained, with each new member fit on residuals of the previous trees. During inference, the weighted sum is taken across the ensemble members. Members are grown or fit by selecting features that provide high information gain (Eq. 2). Information gain is calculated by evaluating the sum of the gradients $G$ and hessians $H$ of the loss function at left and right leaf nodes when selecting features during tree fitting (for our experiments we use Mean Squared Error as the loss function shown in Eq. 3. Features that are informative for reducing residual error during tree development yield high gain values. These values can be summed across trees in the ensemble to produce a ranking of feature contribution. This provides a post-hoc method of interpretability yielding a high level or global view of feature importance to correcting

$\Delta XCO_2$. While this method of interpretability is less informative than the regression coefficients provided by a linear model, it is useful for tasks such as feature selection.

XGBoost employs $L_1$ and $L_2$ norm regularization to reduce overfitting to outliers present in the training dataset. The effect of the regularization is governed by the hyper-parameters $\lambda$ and $\gamma$, and must be carefully selected or tuned. To find these hyper-parameters we use a k-fold cross validation strategy in which the training dataset is divided into $k$ subsets (we use $k$=10) and each subset is sequentially held out for evaluation for a model trained on the rest of the data. Performance across the k-folds is averaged and the process is repeated for each potential selection of hyper-parameters. We found a $\lambda_{LAND}$=2.5 and $\gamma_{LAND}$=3.75 for the land correction, and $\lambda_{OCEAN}$=2.0 and $\gamma_{OCEAN}$=10.0.

$$Information\ Gain = \frac{1}{2}\left[\frac{G_{Left}^2}{H_{Left}+\lambda} + \frac{G_{Right}^2}{H_{Right}+\lambda} - \frac{(G_{Left}+G_{Right})^2}{H_{Left}+H_{Right}+\lambda}\right] - \gamma\ ,$$

(2)

$$Mean\ Squared\ Error\ loss = \frac{1}{N}\sum_{i=1}^{N}(y_i - \hat{y}_i)^2,$$

(3)

We initially evaluated a Random Forest but found better performance with the gradient booster. Therefore, we have focused only on the XGBoost models for demonstrating a non-linear correction in the manuscript.

L136-145: Have you tried other ways to get feature importances from XGBoost, e.g. permutation based importance or importance computed with SHAP values? Is it possible to improve interpretability by a choice tailored to this specific problem?

We have added comparison to permutation feature importance and note good agreement with information gain. We have added a plot and discussion to Appendix A.

**Appendix A: Feature selection and importance**

To assess the robustness of our choice of features, we compare the ranking produced by the information gain feature importance generated by the gradient booster, with the ranking produced by a method called permutation feature importance (Fisher et al. 2018). Permutation feature importance captures the contribution to residual error when a feature has its values randomly shifted across observations. Permutation feature importance is a model agnostic post-hoc method that does not require the bias correction model to be retrained. In Figure A1 we compare the normalized rankings for the individual proxy/surface/mode models that were used to select variables for the final bias correction models trained on all truth proxies. Good agreement is observed in both the overall ranking and magnitude of normalized feature importance between both methods.

[Figure]

**Figure A1. Comparison of feature importance derived from information gain and permutation importance. Normalized importance (permutation importance in stars, and information gain in circles) are shown for land and ocean features, and by truth proxy. Feature importance methods are largely in agreement in ranking and contribution.**

L207-210 and Figure 2: Please list the used features and the respective importances (split by land and ocean) for the models based on the different proxy data sets and for the final bias correction separately in a table.

Figure 2 now shows the ranking and names of features used in the final bias correction.

[revised manuscript text omitted]

L219: Are there different prior surface pressures for the weak and the strong band? If so, why?

There is an alignment offset in the pointing location between the three bands therefore three priors are used.

Table 3: Please remove "large unphysical" in the description of co2_grad_del. Or do you use an additional threshold to rate a deviation from the prior as unphysical?

Removed.

L260: Overfitting of XGBoost cannot be generally excluded. Please prove that there is indeed no overfitting for this specific task and parameter setup. To this end, a more challenging selection of training and validation datasets could also be considered (see also general comments).

We now address this in Section 5.1, shown in the response above to the general comment.

Table 4: Please also list the results for the final proposed bias correction combining all truth proxies.

Added RMSE for combined truth proxies for each model.

Figure 3/4, Table 4/5/6: Only validation data (2018 with truth proxy sampling) is displayed here, right? If this is the case, please be more specific in the description.

All tables and descriptions now thoroughly depict 2018 data.

ADD DESCRIPTION UPDATE HERE

L303-304: Wouldn't it be better to apply the footprint correction before training the bias correction?

This is an excellent point. We now apply the footprint correction first before removing the feature correction. While the effects to performance are negligible, feature contribution to bias is now more accurately depicted. Hyperparameter selection has also been updated.

Is it possible to include the footprint correction in the bias correction by introducing a suitable parameter as feature (e.g. row number)?

This is currently being evaluated for inclusion in a machine learning correction that expands on the approach of this manuscript and scheduled for a L2 litefile update for B11.

Figure 5: Please highlight more (in the text and in the caption) what exactly is shown here: it is all data from 2016-2018 including three types of data (correct me if I'm wrong), namely 1) training data (truth proxy data for 2016-2017), 2) validation data (proxy data for 2018), 3) data beyond (data from

2016-2018 used neither for training nor validation). It would be very enlightening to show (and compare) this kind of figure separately for each of the three data types, because this could provide indications of how well the correction generalises to actually independent data (type 3).

We have now added much needed clarification to the data that is displayed in this plot. In order to increase that data available for plotting we trained three models using with a year of data from 2016 to 2018 used as hold out. The data plotted is the aggregated validation years and would be similarly described as "type 2". Figure 5 text now reads:

**Figure 5. Remaining XCO$_2$ biases ($\Delta$XCO$_2$) after correction for 2016-2018 and model mean proxy, binned to a 2°x2° resolution. $\Delta$XCO$_2$ after the XGBoost correction for QF=0 is shown in (a), $\Delta$XCO$_2$ after the B10 correction for QF=0 is shown in (b), $\Delta$XCO$_2$ after the XGBoost correction for QF=1 is shown in (c), $\Delta$XCO$_2$ after the B10 correction for QF=1 is shown in (d), and difference (B10 – XGB) for QF=0 is shown in (e). Three models are trained each with one year in [2016,2017,2018] used as holdout. The results on the holdout sets are then used for plotting.**

Table 6: It would be interesting to add the respective performances for type-3-data (or to introduce more challenging training and validation data sets from the start, see also general comments). Due to the lack of an entirely independent validation dataset, the performance suggested here may be too optimistic.

We now address "type 3" data in the new section 5.1 as described above in the response to the general comment.

Figure 7/8: Please explain all occurring variables.

We have added appendix Table C1 that includes all variables used or considered in either bias correction or filtering.

[revised manuscript text omitted]

Section 5.1, Figure 9: Please also report the individual information gains and not only the differences in feature importances.

Figure 9 now shows the individual information gain. Wording the section has also been changed to reflect this.

**5.2 Evaluating feature importance between filter regimes**

To understand the contribution of the features to correcting bias in QF=0 and QF=1 data, we compare the information gain between the two regimes. To perform the ablation study we again employ the models trained on individual truth proxies and re-train and evaluate them on QF=0 and again for QF=1 data. Figure 9 shows the information gain for each filter regime for land and for ocean. For land, dpfrac and co2_grad_del are highly informative for correction of QF=0 data by the machine learning model. Similarly for ocean QF=0 data, the surface pressure delta term dp_sco2 and co2_grad_del are also highly informative. In operation, these terms are also used for bias correction in all ACOS versions (dpfrac replaced dP in B10) to date. These variables are responsible for the largest reduction in unexplained variance in the filtered regime (Payne et al. 2022; Osterman et al. 2020; O'Dell et al. 2018)

For land QF=1 data, there is a drop in importance for co2_grad_del and dpfrac and large increase for h2o_ratio and relative increases for the albedo and aerosol terms. To explain the high importance for the h2o_ratio, we look to the non-linear interaction outside of the bound imposed by the operational filter which removes soundings with a h2o_ratio greater than 1.023, reducing the regime of interaction to one that is not highly correlated with $\Delta XCO_2$. In the QF=1 regime, h2o_ratio corresponds to a significant negative bias. Larger values of h2o_ratio are explained in Taylor et al. 2016, where it was shown that retrieved surface albedo from the strong $CO_2$ band is generally lower than the weak $CO_2$ band. In cases of larger aerosol presence, this sensitivity leads to weaking of the absorption features and a positive departure from unity. The additional albedo term for the strong $CO_2$ band as well as the additional aerosol terms also increase in importance for QF=1.

For ocean QF=1 data, there is a significant change in information gain for several features. The surface pressure delta term dp_sco2, becomes significantly less informative for correcting QF=1

where negative values of dp_sco2 are relatively uncorrelated with $\Delta XCO_2$. Similarly, to land, the albedo term for the strong $CO_2$ band more informative for correcting outside the filtered regime along with the residual error between forward modelled radiances and measurements in the weak $CO_2$ band.

[Figure]

**Figure 9: Feature importance for land is shown in (a), feature importance for ocean is shown in (b). Y-axis displays the normalized information gain from XGBoost models with QF=0 shown in darker colours and QF=1 shown in lighter colours.**

Technical Corrections

There are incorrect Figure and Table numbers used in the main text. Please check.

Corrected.

Citations in the text do not always match the ones in the References section. Please check.

Thank you for catching these, we have corrected several citation errors.

L12: "Obersvatory" → "Observatory"

Corrected.

L15: "correlate" → "correlated"

Corrected.

L35: Please remove the closing bracket.

Corrected.

L36: It is (Rodgers, 2000).

Corrected.

L47: "epmerically". Do you mean "empirically"?

Corrected.

L49: "inturn" → "in turn"

Corrected.

L53: "reduce" → "reduces"

Corrected.

L55: "applying quality filter" → "applying the quality filter"

Corrected.

L55: Please remove the full stop between "correction" and "to" or rephrase both sentences.

Corrected

Figure 1: The number of soundings N is wrong in panel (a). The colour scale is oversaturated. Please extend the value range (0-2000 does not seen to be optimal).

Corrected

[Figure]

**Figure 1: Spatial coverage for each truth proxy. The mean of a set of flux models is shown in (a), small area approximation is shown in (b), and TCCON is shown in (c).**

L106: "overs" → "offers"

Corrected

L178: "trainined" → "trained"

Corrected

L186: "This, allows" → "This allows"

Corrected

L223: "retrivals" → "retrievals"

Corrected

L252: The section numbering is inconsistent. This number already exists.

Corrected

This list of technical corrections is (likely) not exhaustive. Please check the text for further typing errors

---

## Author Response (AR2)

**A non-linear data driven approach to bias correction of XCO2 for OCO-2 NASA ACOS version 10**

**William Keely et al.**

The points from the reviewer are shown in black, with our response in blue, and changes or additions to the manuscript in red.

**Response to Reviewer #1**

Specific Comments:

The authors correctly state that features measuring the deviation of retrieved quantities from the prior (surface pressure and vertical $CO_2$ profile) have already been used in the operational bias correction. However, with a much more complex non-linear machine learning method, it actually needs to be checked again that there is no attenuation of actual $CO_2$ signals when using these (or other) features in the bias correction. In section 5.3, two examples are used to demonstrate that the machine learning corrected product captures enhancements not present in the training data. This is a good thing. However, one objective of the non-linear correction was to largely reproduce the linear model for QF=0 (which is the currently accepted community standard) and Figure 6 additionally shows that the results associated to XGBoost_QFNew and B10_QF are very similar in terms of sounding throughput for both analysed regions (South Korea and Ohio, US). It is therefore reasonable to assume that most of the soundings in Figure 10 are of type QF=0 and that these examples are therefore unlikely to reflect the main innovations of QFNew quality filtering. The probably more critical part of the proposed non-linear bias correction, namely the increase of sounding throughput beyond QF=0 and the behaviour for related soundings with QF=1 in terms of potential over-correction, is thus (most likely) not explicitly investigated. This would be a worthwhile addition. Furthermore, Figure 9 shows that the specific features measuring the deviation of retrieved quantities from the prior are also important for QF=1. Thus, the following questions arise, whose answers would significantly further increase the conclusiveness of Section 5.3:

Are most of the soundings shown in Figure 10 actually of type QF=0?
How many soundings (absolute and relative) in Figure 10 are of type QF=1?
Can you introduce a graphical distinction of soundings with QF=0 and QF=1 in Figure 10?
Is it possible to find a plume, which is not present in the training data and largely consists of QF=1 data?

This is an excellent point, and we agree with the need for further assessment in Section 5.3. While the US plume is indeed primarily QF=0 data, the South Korea plume is primarily QF=1 (~65%) and we believe is suitable for the analysis. QFNew passes ~25% of QF=1 data for Taean, S. Korea much of which is within the plume feature itself. We have added an additional plot to Figure 10:

[Figure]

**Figure 10. Two CO₂ plumes captured downwind from power plants (Nassar et al. 2021). An ocean glint and land nadir plume at Taean, South Korea, [lat 36.91°, lon 126.23°] on 2015-04-17 is shown in (a). A land nadir plume near the J. M. Gavin and Kyger Creek power plants in Ohio, USA, [lat 38.93°, lon -82.12°] on 2015-07-30. Regions with the example plumes are not present in the training dataset and consist of QF = 0 + 1 data. Plot (c) shows the increase in XGBoost corrected data for QFNew=0 that would be filtered by the B10 QF.**

Please elaborate in the manuscript on the difference QFNew\B10_QF of the two sets QFNew and B10_QF (set of elements of QFNew not in B10_QF) in the context of potentially correcting out actual CO2 signals.

We have updated both 5.3 and 5.4 in the manuscript.

**5.3 Preservation of CO₂ enhancements**

We assess the risk of the proposed bias correction to correct out and remove plume features in the data. Several features heavily utilized by the XGBoost models and in operational correction such as the CO₂ gradient delta, and surface pressure terms (e.g., dpfrac, dp_o2a), are differences between the ACOS retrieved state, and the prior. Therefore, there is potentially a risk for the bias correction to use the delta terms to over correct the retrieved XCO₂ to the truth. We compare XGBoost corrected XCO₂ for two known plumes first identified in Nassar et al. 2021. The two example plumes are shown in Figure 10 (a) and (b): an ocean glint and land nadir plume in Taean, South Korea, and a land nadir plume observed over two co-located power plants in Ohio, US. We compare the uncorrected XCO₂ retrieval (B10 Raw), the operationally corrected XCO₂ (B10 Corrected) and the machine learning corrected XCO₂ (XGBoost Corrected) and note that the machine learning corrected product captures enhancements not present in the training data. These results are also consistent with the findings in Mauceri et al. 2023 which include similar delta terms. This is further illustrated with the Taean plume which consists of ~35% QF = 0 soundings and ~65 QF = 1 soundings. QFNew = 0 improves the passing rate to ~ 60% as shown in Figure 10 (c). The red stars show data that is passed by QF = 0 (and by construction QFNew = 0) and the blue stars show data that would be removed by QF = 1 but is passed by QFNew = 0, indicating where the increase of available data for the plume feature. Of particular interest is the increase of data within the feature around 36.95° which includes maximum observed enhancement value.

**5.4 Potential for further improving data throughput**

Figure 11 further illustrates how the shape of the filtering or decision surface can affect data throughput. Soundings are binned by two state vector features: h2o_ratio and dpfrac. Figure 11b, and Figure 11d show the improvement in reduction of mean $\Delta XCO_2$ and in the error divided by the posterior uncertainty, from the non-linear correction. The QF filters for each feature are indicated by the black dashed lines and the interior of the intersection of these filters indicates the region of state space that is labelled as QF = 0 (Note: the additional filters of the QF further reduce the data that is passed in this region). Significant portions of the distribution, where the non-linear method can accurately correct, lay outside of this filtered region and are labelled QF = 1. A data driven filter can be constructed using similar interpretable machine learning techniques and produce a unified correction/filtering product. Furthermore, moving away from the binary quality flag to a ternary ("very good", "good", "bad") will likely provide an improved data product for end users. Data driven methods for quality filtering have already proven to be useful in the northern high latitudes (Mendonca et al. 2021) and a genetic algorithm was previously used to derive the Warn Levels which complement the operational quality flag found in early OCO-2 data versions (Mandrake et al. 2015). An important task for such future work will be to ensure that the machine learning method learns a physically consistent filter that can increase data throughput while still limiting variance of error and $\Delta XCO_2$. We also acknowledge that while the Taean plume shown in Figure 10 illustrates an empirical example of the ability of a non-linear correction to improve throughput of good quality data, further evaluation of the intersection (QF = 1 & QFNew = 0) will be required before bringing such a method to operation.

Technical Corrections:

L28: Bovensmann
Corrected.
L43: under-constrained
Corrected
L65: provides
Corrected
L128-131: Please make it two sentences
Corrected
L191: train a set
Corrected
L261: There was already a Section 4.1 before. Please correct the section numbering
Corrected
L383: becomes
Corrected

**Response to Reviewer #2**

Minor

Figure 5/Table 7: It would be ideal to use only the 2018 evaluation data here. I understand the point the authors make about needing more data to plot (and thus switching to the average of three models for three years), but this is inconsistent with the rest of the paper. This is in theory fine for Figure 5 because it is explained, but it is not okay for Table 7 where the claim is made that the RMSE of the B10 correction is maintained. This is because Table 7 needs to be compared to Table 4 to see the RMSE has been maintained and Table 4 only evaluates on 2018 data. I suggest the authors use just 2018 data to be consistent with the rest of the paper.

Figure 5, Figure 6, and Table 7 now only show results from 2018.

[Figure]

**Figure 5. Remaining XCO₂ biases (ΔXCO₂) after correction for 2018 and model mean proxy, binned to a 3º×3º resolution. ΔXCO₂ after the XGBoost correction for QF=0 is shown in (a), ΔXCO₂ after the B10 correction for QF=0 is shown in (b), ΔXCO₂**

after the XGBoost correction for QF=1 is shown in (c), ΔXCO$_2$ after the B10 correction for QF=1 is shown in (d), and difference (B10 − XGB) for QF=0 is shown in (e).

**Table 7. RMSE for combined XGBoost correction, B10 QF percent data throughput, and QFNew percent data throughput, by surface/mode, for 2018.**

| Surface (Mode) | XGBoost RMSE | B10 % Passing | QFNew % Passing |
|---|---|---|---|
| Land (Nadir+Glint) | 1.07 ppm | 59% | 69% |
| Ocean (Glint) | 0.72 ppm | 60% | 74% |

[Figure]

**Figure 6. Relative increase in percent passing QFNew over B10 QF for 2018 aggregated by 4º×4º bins.**

Technical

Line 31: Orbital -> Orbiting
Line 44: remove ")"
Line 63: rephrase sentence starting "A drawback of applying…"
Line 65: or is -> and are (?)
Line 86: 2022 -> 2023
Line 96: kernel -> averaging kernel
Line 106: is still -> it is still
Line 131: co-located TCCON -> co-located with TCCON
Line 191: run on sentence. Maybe it is meant to split between "nodes" and "when" (?)
Line 272: It is mentioned earlier that data is available until February 2019. Is January 2019 to February 2019 ignored? This should be specified.
Line 302: train as -> train a
Line 314: correction to -> correction
Line 352: for operational -> for the operational

Line 354: missing words before "and prior" (?)
Figure 2: make x-axis labels consistent with Table 3
Line 560: repeated section number (not corrected as noted in response to reviewers)
Lines 565-566: recalculate numbers in bold
Section 4.2 -> clarify that you are using the XGBoost (trained on QF = 0 + 1)
Line 602: 1.37 -> 1.38
Line 826: order of y vs. x is flipped
Line 951: add "et al."
Line 1024: appears to be an incomplete sentence (maybe a former figure caption?)
Line 1027: sentence that begins "likely" is incomplete
Line 1045/1054: QF -> QFNew
Line 1046: spelling of parentheses
Line 1053: diamonds -> pluses
Table C1: define abbreviations in third column; co2_grad_del description is not consistent with Table 3 ("large unphysical"), define what the asterisk means.

Thank you for thoroughly reviewing and catching these errors. We have corrected for these and additional mistakes throughout the manuscript.